# Mutations that improve efficiency of a weak-link enzyme are rare compared to adaptive mutations elsewhere in the genome

Andrew B Morgenthaler[1,2], Wallis R Kinney[1,2], Christopher C Ebmeier[1], Corinne M Walsh[2,3], Daniel J Snyder[4], Vaughn S Cooper[4], William M Old[1], Shelley D Copley[1,2]*

[1]Department of Molecular, Cellular, and Developmental Biology, University of Colorado Boulder, Boulder, United States; [2]Cooperative Institute for Research in Environmental Sciences, University of Colorado Boulder, Boulder, United States; [3]Department of Ecology and Evolutionary Biology, University of Colorado Boulder, Boulder, United States; [4]Center for Evolutionary Biology and Medicine, University of Pittsburgh, Pittsburgh, United States

**Abstract** New enzymes often evolve by gene amplification and divergence. Previous experimental studies have followed the evolutionary trajectory of an amplified gene, but have not considered mutations elsewhere in the genome when fitness is limited by an evolving gene. We have evolved a strain of *Escherichia coli* in which a secondary promiscuous activity has been recruited to serve an essential function. The gene encoding the 'weak-link' enzyme amplified in all eight populations, but mutations improving the newly needed activity occurred in only one. Most adaptive mutations occurred elsewhere in the genome. Some mutations increase expression of the enzyme upstream of the weak-link enzyme, pushing material through the dysfunctional metabolic pathway. Others enhance production of a co-substrate for a downstream enzyme, thereby pulling material through the pathway. Most of these latter mutations are detrimental in wild-type *E. coli,* and thus would require reversion or compensation once a sufficient new activity has evolved.

*For correspondence:
Shelley.Copley@Colorado.EDU

**Competing interests:** The authors declare that no competing interests exist.

## Introduction

The expansion of huge superfamilies of enzymes, transcriptional regulators, transporters, and signaling molecules from single ancestral genes has been a dominant process in the evolution of life (*Bergthorsson et al., 2007*; *Chothia et al., 2003*; *Glasner et al., 2006*; *Hughes, 1994*; *Ohno, 1970*; *Todd et al., 2001*). The emergence of new protein family members has enabled organisms to access new nutrients, sense new stimuli, and respond to changing conditions with ever more sophistication (*Conant and Wolfe, 2008*; *Nei and Rooney, 2005*; *Reams and Neidle, 2004*; *Santos et al., 2017*; *Starr et al., 2017*; *Storz, 2016*).

The Innovation-Amplification-Divergence (IAD) model (*Figure 1*) posits that evolution of new enzymes by gene duplication and divergence begins when a physiologically irrelevant promiscuous activity becomes important for fitness due to a mutation or environmental change (*Bergthorsson et al., 2007*; *Francino, 2005*; *Hughes, 1994*; *Näsvall et al., 2012*). A newly useful enzymatic activity is often inefficient, making the enzyme the 'weak-link' in metabolism. Gene duplication/amplification provides a ready mechanism to improve fitness by increasing the abundance of a weak-link enzyme. If mutations lead to an enzyme capable of efficiently carrying out the newly needed function, selective pressure to maintain a high copy number will be removed, allowing extra

**Figure 1.** The Innovation-Amplification-Divergence (IAD) model of gene evolution. A promiscuous activity B of an enzyme may become physiologically relevant due to a mutation or environmental change. Gene amplification increases the abundance of the weak-link enzyme. Mutations can improve the efficiency of the newly important activity B. Once sufficient B activity is achieved, selection is relaxed and extra gene copies are lost, leaving behind two paralogs.

copies to be lost and leaving behind two paralogs (or just one gene encoding a new enzyme if the original function is no longer needed).

While the IAD model provides a satisfying theoretical framework for the process of gene duplication and divergence, our understanding of the process is far from perfect. Although the signatures of gene duplication and divergence are obvious in extant genomes, we have little information about the genome contexts and environments in which new enzymes arose. Laboratory evolution offers the possibility of tracking this process in real time. In a landmark study, Näsvall et al. used laboratory evolution to demonstrate that a gene encoding an enzyme with two inefficient activities required for synthesis of histidine and tryptophan amplified and diverged to alleles encoding two specialists within 2000 generations (*Näsvall et al., 2012*; *Newton et al., 2017*). However, this study followed only mutations in the diverging gene. When an organism is exposed to a novel selection pressure that requires evolution of a new enzyme, any mutation – either in the gene encoding the weak-link enzyme or elsewhere in the genome – that improves fitness will provide a selective advantage.

We have explored the relative importance of mutations in a gene encoding a weak-link enzyme and elsewhere in the genome using a model system in *Escherichia coli*. ProA (γ-glutamyl phosphate reductase, *Figure 2*) is essential for proline synthesis in *E. coli*. ArgC (*N*-acetylglutamyl phosphate reductase) catalyzes a similar reaction in the arginine synthesis pathway, although the two enzymes are not homologous (*Goto et al., 2003*; *Ludovice et al., 1992*; *Page et al., 2003*). ProA can reduce *N*-acetylglutamyl phosphate (NAGP), but its activity is too inefficient to support growth of a Δ*argC* strain of *E. coli* in glucose. However, a point mutation that changes Glu383 to Ala allows slow growth of the Δ*argC* strain in glucose. Enzymatic assays show that E383A ProA (ProA*) has severely reduced activity with γ-glutamyl semialdehyde (GSA), but substantially improved activity with *N*-acetylglutamyl semialdehyde (NAGSA) (*Khanal et al., 2015*; *McLoughlin and Copley, 2008*). (It is necessary to assay kinetic parameters in the reverse direction because the substrates for the forward reaction are too unstable to prepare and purify.) Glu383 is in the active site of the enzyme; the change to Ala may create extra room to accommodate the larger substrate for ArgC, but at a cost to the ability to bind and orient the native substrate. The poor efficiency of the weak-link ProA* creates strong

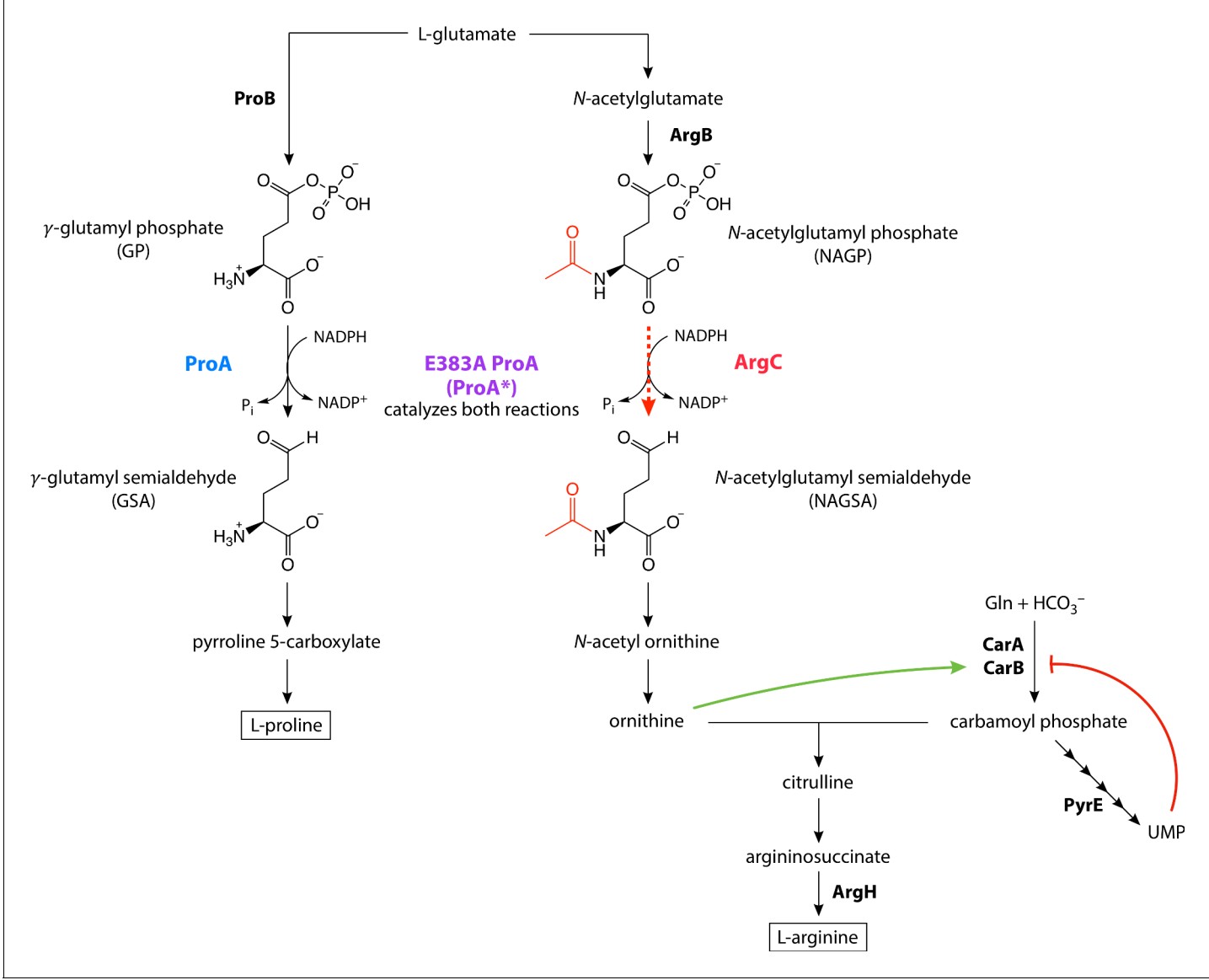

**Figure 2.** E383A ProA (ProA*) replaces ArgC in the arginine synthesis pathway in Δ*argC proA\* E. coli*, but is the bottleneck in the pathway due to its poor catalytic activity. The reaction normally catalyzed by ArgC and replaced by ProA* in the parental strain is indicated by the red dotted line. The green and red lines indicate allosteric activation and inhibition, respectively.

The online version of this article includes the following figure supplement(s) for figure 2:

**Figure supplement 1.** Genome modifications in the parental AM187 strain.

selective pressure for improvement of both proline and arginine synthesis during growth of Δ*argC E. coli* on glucose as a sole carbon source.

We evolved eight replicate populations of Δ*argC proA\* E. coli* in minimal medium supplemented with glucose and proline for up to 1000 generations to identify mechanisms by which the impairment in arginine synthesis could be alleviated. Our expectation that amplification of *proA\** would be beneficial was borne out in all populations. Whole-genome sequencing of the adapted populations and further biochemical analysis showed that an adaptive mutation in *proA\** followed by deamplification of *proA\** occurred in only one population. Indeed, most of the adaptive mutations occurred outside of *proA\**. We have identified the mechanisms by which three common classes of such mutations increase fitness: (1) restoration of a known defect in pyrimidine synthesis; (2) an increase in the amount of ArgB, the enzyme that synthesizes NAGP, the substrate for the weak-link ProA*; and (3)

an increase in flux through carbamoyl phosphate synthetase, whose product feeds into the arginine synthesis pathway downstream of the weak-link enzyme (*Figure 2*). The latter two types of mutations appear to increase flux through the bottlenecked arginine synthesis pathway while the more difficult process of improving the weak-link enzyme progresses. In the case of the mutations affecting carbamoyl phosphate synthetase, the fitness increase comes at a cost to presumably well-evolved regulatory functions.

Our results demonstrate that mutations elsewhere in the genome play an important role during the process of gene amplification and divergence when the inefficient activity of a weak-link enzyme limits fitness. Thus, the process of evolution of a new enzyme by gene duplication and divergence is inextricably intertwined with mutations elsewhere in the genome that improve fitness by different mechanisms.

## Results

### Growth rate of Δ*argC proA\* E. coli* increased 3-fold within a few hundred generations of evolution in M9/glucose/proline

We generated a progenitor strain for laboratory evolution by replacing *argC* with the *kan^r* antibiotic resistance gene, modifying *proA* to encode ProA\*, and introducing a mutation in the −10 region of the promoter of the *proBA* operon. (This mutation was one of two promoter mutations previously shown to increase *proA\** expression during adaptation of the Δ*argC* strain [*Figure 2—figure supplement 1*; *Kershner et al., 2016*]). The presence of the promoter mutation ensured that all populations had the same mutation during the evolution experiment. We also introduced *yfp* downstream of *proA\** and deleted several genes (*fimAICDFGH* and *csgBAC*, which are required for the formation of fimbriae and curli, respectively *Barnhart and Chapman, 2006*; *Proft and Baker, 2009*) to minimize the occurrence of biofilms. We evolved eight parallel lineages of this strain (AM187, *Table 1*) in M9 minimal medium supplemented with 0.2% (w/v) glucose, 0.4 mM proline, and 20 μg/mL kanamycin in a turbidostat to identify mutations that improve arginine synthesis. We used a turbidostat rather than a serial transfer protocol because turbidostats can maintain cultures in exponential phase and thereby avoid selection for mutations that simply decrease lag phase or improve survival in stationary phase. Turbidostats also avoid population bottlenecks during serial passaging that can result in loss of genetic diversity.

Growth rate in each culture tube was averaged over each 24 hr period and was used to calculate the number of generations each day. Each culture was maintained until a biofilm formed (33–57 days, corresponding to 470–1000 generations). While it is possible to restart cultures from individual clones after biofilm formation, this practice introduces a severe population bottleneck. Thus, we decided to stop the evolution for each population when a biofilm formed.

Over the course of the experiment, growth rate increased 2.5–3.5-fold for all eight populations (*Figure 3*). Occasional dips in growth rate occurred during the evolution. These dips are artifacts arising from temporary aberrations in selective conditions due to turbidostat malfunctions that prevented introduction of fresh medium, causing the cultures to enter stationary phase. Occasionally cultures were saved as frozen stocks until the turbidostat was fixed (see Materials and methods). Restarting cultures from frozen stocks may have caused a temporary drop in growth rate.

### Copy number of *proA\** and size of the amplified genomic region varied among replicate populations

We monitored *proA\** copy number during the evolution experiment using qPCR of population genomic DNA (*Figure 4A*, *Figure 4—figure supplement 1*). *proA\** was present in at least six copies by generation 300 in all eight populations. Six of the populations maintained 6–9 copies for the remainder of the adaptation. *proA\** copy number in population 2 increased to as many as 20 copies. In population 3, *proA\** copy number dropped to three by generation 400.

We identified the boundaries of the amplified regions in all eight populations by sequencing population genomic DNA (*Figure 4B*, *Figure 4—source data 1*). The amplified region in population 2 was unusually small, spanning only 4.9 kb and resulting in co-amplification of only two other genes besides *proBA\**. Population 2 also appeared to have a second region of amplification of 18.5 kb. (Whether these two distinct amplification regions coexisted in the same clone or as two separate

**Table 1.** Strains used in this work.

| strain | genotype | notes |
|---|---|---|
| | *E. coli* BW25113 | GenBank accession number CP009273 (*Grenier et al., 2014*) |
| AM008 | *E. coli* BW25113; *argC::kan*[r] | Keio strain (*Baba et al., 2006*) |
| AM187 | AM008 + −45 C→T upstream of *proB* (M2 promoter mutation from *Kershner et al., 2016*) + A1148→C in *proA* (changes Glu383 to Ala) + *yfp* construct downstream of *proBA* consisting of (from 5′to 3′) BBa_B0015 terminator, P3 promoter, synthetic RBS, *yfp* (see Materials and methods); ΔfimAICDFGH; ΔcsgBAC | parental strain for adaptation, GenBank accession number CP037857.1 |
| AM209 | *E. coli* BL21(DE3); *argC::kan*[r]; *proA::cat* | used for expression and purification of wild-type and mutant ProAs |
| AM239 | AM187 + 58 bp deletion upstream of *argB* (pos. 4145856–4145913)[a] | |
| AM241 | AM187 + C4145901→G (24 bp upstream of *argB* start codon) | |
| AM242 | AM187 + C4145903→A (22 bp upstream of *argB* start codon) | |
| AM243 | AM187 + C4145903→T (22 bp upstream of *argB* start codon) | |
| AM244 | AM187 + C4145907→A (18 bp upstream of *argB* start codon) | |
| AM245 | AM187 + 38 bp duplication upstream of *argB* (pos. 4145912–4145949) | |
| AM267 | *E. coli* BL21; *carAB::kan*[r] | used for expression and purification of wild-type and mutant carbamoyl phosphate synthetases |
| AM279 | AM187 + C1169→T in *ygcB* (changes Ala390 to Val in Cas3) | |
| AM320 | AM187 + T1116→G in *proA** (changes Phe372 to Leu) | |
| AM327 | AM187 + 82 bp deletion upstream of *pyrE* (pos. 3808881–3808962) | |
| AM329 | AM187 + 82 bp deletion upstream of *pyrE* (pos. 3808881–3808962); C1169→T *ygcB* (changes Ala390 to Val in Cas3) | |
| AM399 | AM187 + Δ2906–2917 *carB* | |
| AM401 | AM187 + Δ2986–3117 *carB* | |
| AM407 | AM187 + *kan*[r]*::argC(null)* (Gly153 and Cys154 changed to TAA stop codons) | |
| AM413 | AM187 + G1106→T *carB* (changes Gly369 to Val) | |
| AM415 | AM187 + A2896→G *carB* (changes Lys966 to Glu) | |
| AM437 | AM187 + C1148→A in *proA** (reverts E383A mutation) | |
| AM439 | AM320 + C1148→A in *proA*** (reverts E383A mutation) | |
| AM441 | *E. coli* BW25113 + 82 bp deletion upstream of *pyrE* (pos. 3808881–3808962) | |
| AM443 | AM441 + Δ2906–2917 *carB* | |

*Table 1 continued on next page*

*Table 1 continued*

| strain | genotype | notes |
|--------|----------|-------|
| AM445 | AM441 + Δ2986–3117 *carB* | |
| AM447 | AM441 + G1106→T *carB* (changes Gly369 to Val) | |
| AM449 | AM441 + A2896→G *carB* (changes Lys966 to Glu) | |

[a] Genome positions refer to the sequence of strain AM187 (GenBank accession number CP037857), which was modified from the *E. coli* BW25113 sequence (GenBank accession number CP009273; **Grenier et al., 2014**) based on the mutations that had been introduced.

clades within the population could not be determined from population genome sequencing.) In contrast, the amplified regions in the other seven populations ranged from 41.1 to 163.8 kb, encompassing between 55 and 177 genes. We attribute the variation in *proA\** copy number to these differences in the size of the amplified region on the genome. The population with the smallest amplified region (4.9 kb, population 2) carries fewer multicopy genes and thus should incur a lower fitness cost, allowing *proA\** to reach a higher copy number (**Adler et al., 2014**; **Kugelberg et al., 2006**; **Pettersson et al., 2009**; **Reams et al., 2010**).

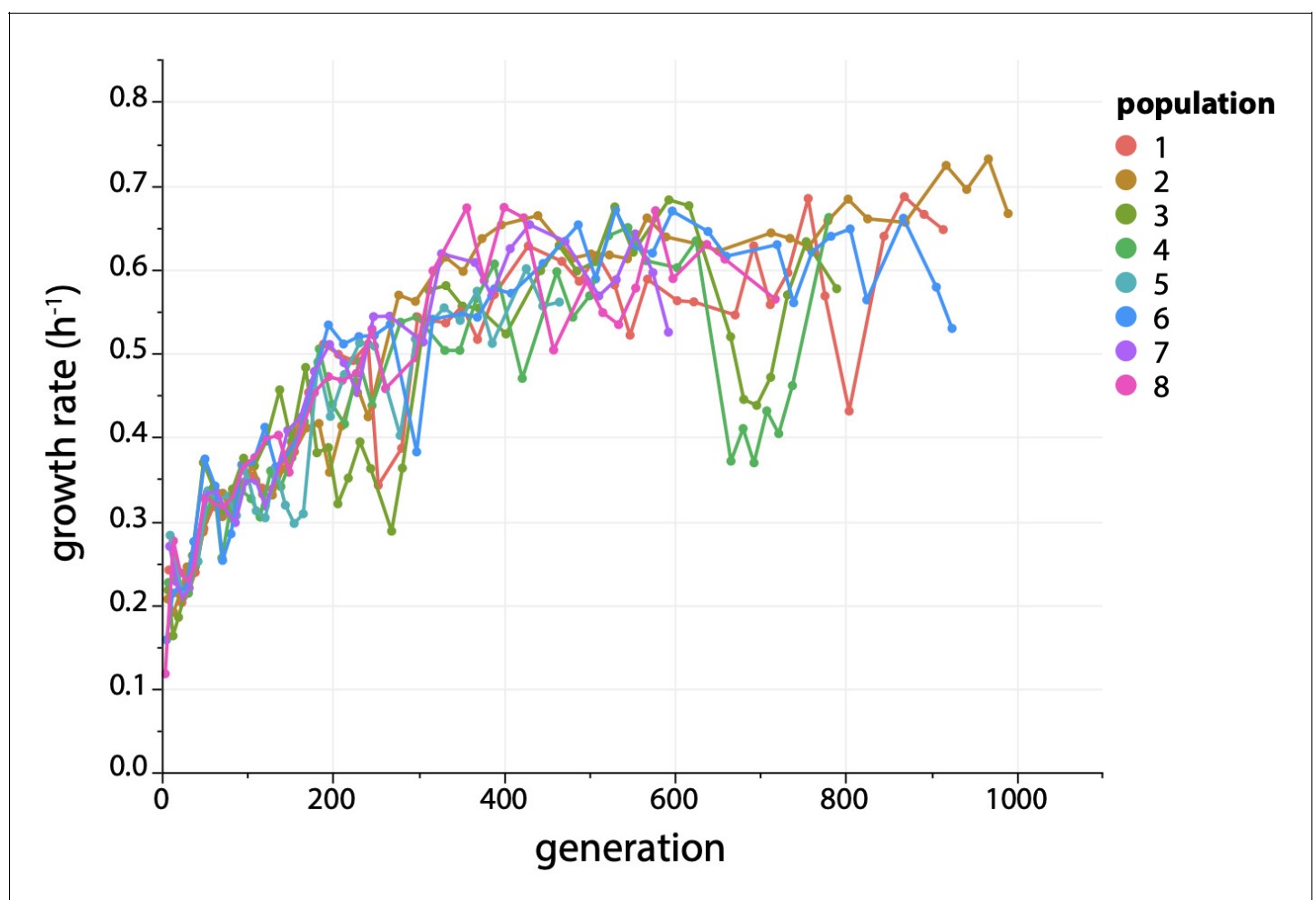

**Figure 3.** Growth rate increases ~ 3 fold during evolution of Δ*argC* M2-*proA\* E. coli* in M9 minimal medium containing 0.2% glucose (w/v), 0.4 mM proline and 20 µg/mL kanamycin. M2 is the C to T mutation at −45 in the promoter for the *proBA* operon (**Kershner et al., 2016**).

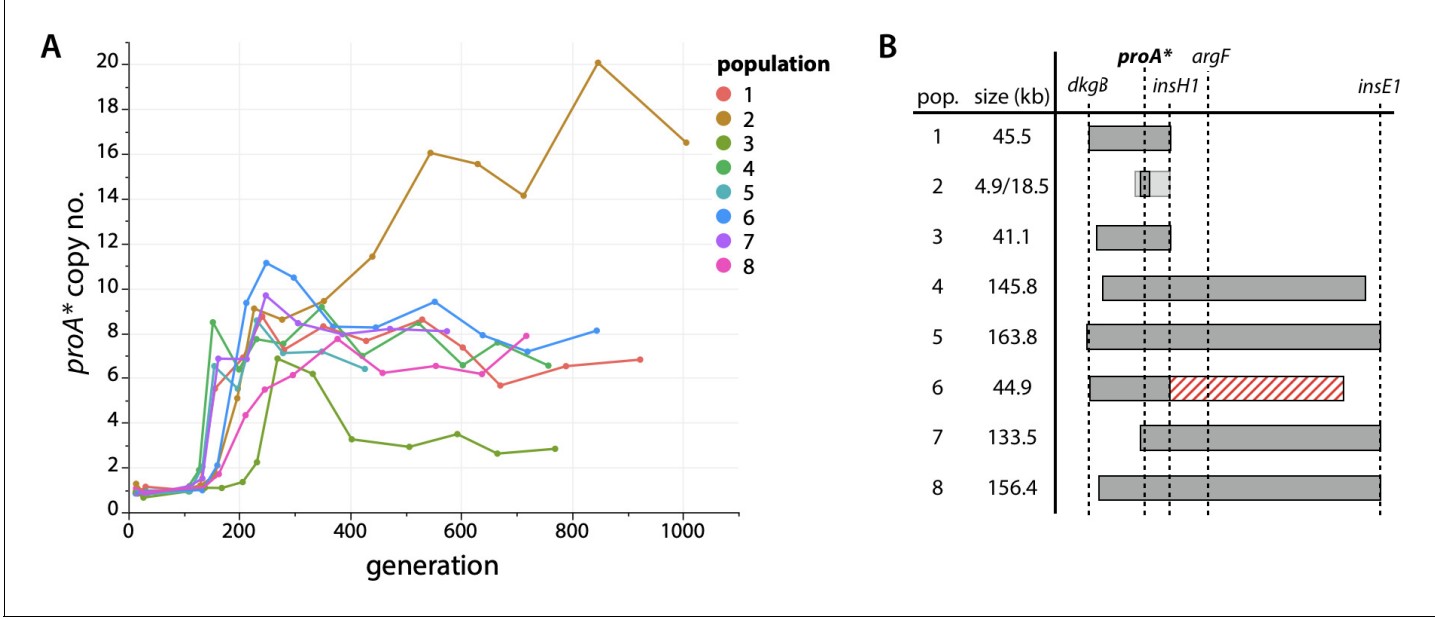

**Figure 4.** *proA\** is amplified during evolution. (A) *proA\** copy number in each evolved population as measured by qPCR. (B) Regions of amplification in each evolved population based on population genome sequencing. Population 2 had two overlapping regions of amplification, both of which included *proA\** (shown as differently shaded bars). Population 6 had a 95.1 kb deletion (shown as a red striped bar) immediately downstream of the amplified region.

The online version of this article includes the following source data and figure supplement(s) for figure 4:

**Source data 1.** Mutations found during the evolution experiment.

**Figure supplement 1.** Growth rate (right axis, dotted lines) and *proA\** copy number (left axis, solid lines) for each evolved population.

**Figure supplement 2.** Effects of the A390V Cas3 and *rph-pyrE* mutations on growth rate of the parental strain AM187.

## A mutation in *proA\** led to deamplification in population 3

The decrease in *proA\** copy number in population 3 was noteworthy since it might have been an indication that a mutation had improved the neo-ArgC activity of ProA\*, resulting in a decreased need for multiple copies. In fact, a mutation in *proA\** that changes Phe372 to Leu (*Figure 5A*) was observed in population 3. E383A F372L ProA will be designated ProA\*\* hereafter. Introduction of this mutation into the parental strain (which carried *proA\**) increased growth rate by 75% (*Figure 5B*), confirming that the mutation is adaptive. Notably, no mutations in *proA\** were identified in any of the other populations.

To determine whether the beneficial effect of the F372L mutation depended upon the presence of the initial E383A mutation, we created variants of the parental strain with either wild-type ProA, F372L ProA, E383A ProA (ProA\*), or F372L E383A ProA (ProA\*\*) (*Figure 5—figure supplement 1*). Strains with either wild-type or F372L ProA did not grow after eight days. Thus, the F372L mutation is not beneficial on its own, and the combined effect of the two mutations is greater than the sum of their individual effects.

The neo-ArgC and native ProA activities of wild-type, ProA\*, and ProA\*\* were assayed (in the reverse direction) with NAGSA and GSA, respectively (*Table 2*). The $k_{cat}/K_{M,NAGSA}$ for ProA\*\* is 3.6-fold higher than that of ProA\* and nearly 80-fold higher than that for ProA. In contrast, there is no difference between $k_{cat}/K_{M,GSA}$ for ProA\* and ProA\*\*.

To determine when the mutation that changes Phe372 to Leu in ProA\* occurred, we sequenced population genomic DNA at generations 270, 440, and 630 and at the end of the evolution (*Figure 5C*). *proA\*\** was present in 9% of the sequencing reads by generation 270. By the time deamplification of *proA\** had occurred at generation 440, the frequency of *proA\*\** had risen to 21% of sequencing reads. By the end of the adaptation, *proA\*\** was fixed in the population, yet three copies remained in the genome, suggesting that ProA\*\* does not have sufficient neo-ArgC activity to be present at a single copy in the genome.

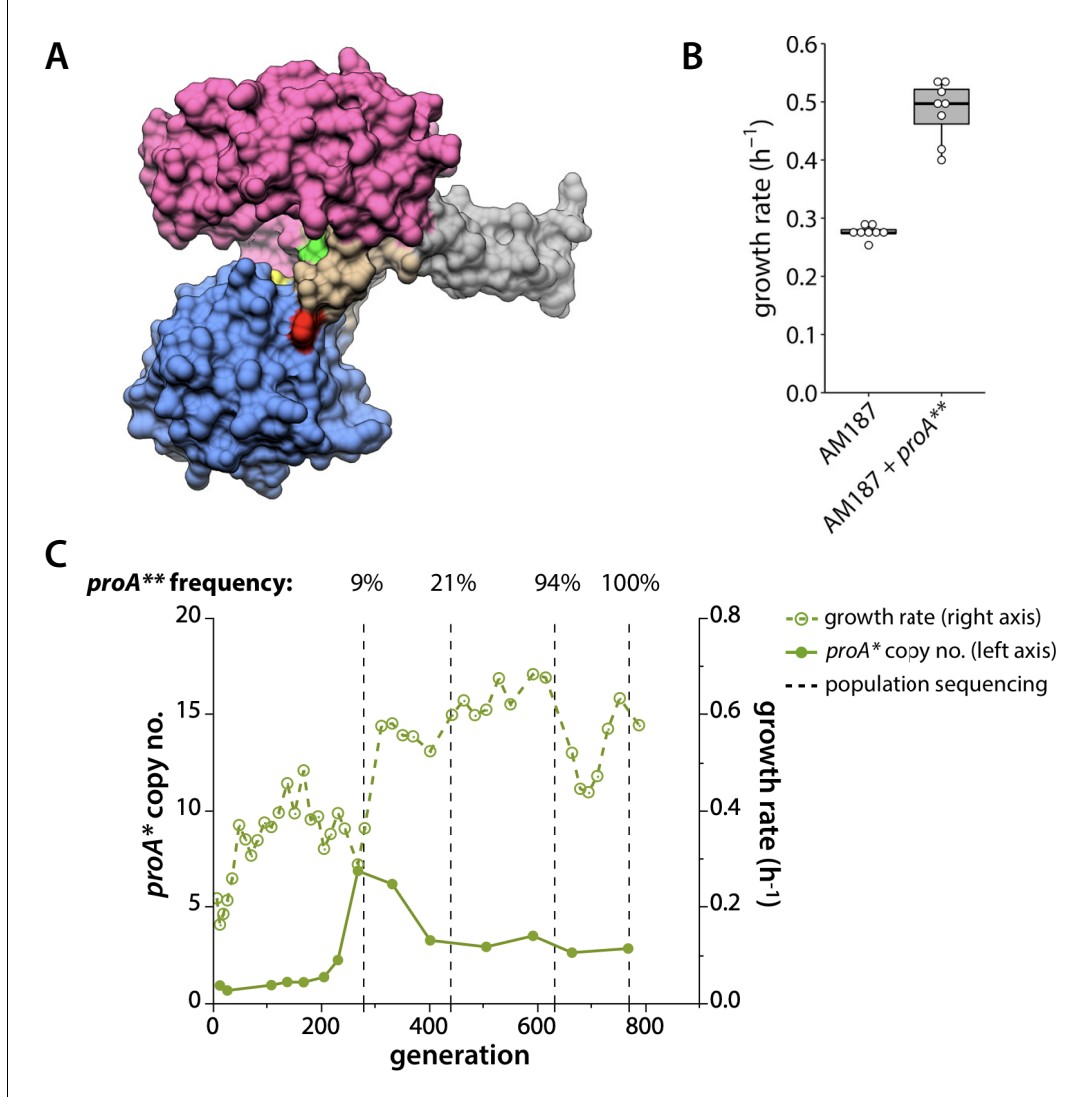

**Figure 5.** *proA** acquired a beneficial mutation in population 3. (**A**) Crystal structure of *Thermotoga maritima* ProA (PDB 1O20) (*Page et al., 2003*). Yellow, catalytic cysteine; green, equivalent of *E. coli* ProA Glu383; red, equivalent of *E. coli* ProA Phe372; magenta, NADPH-binding domain; blue, catalytic domain; beige, hinge region; gray, oligomerization domain. (**B**) Change in growth rate when the mutation changing Phe372 to Leu (*proA***) is introduced into the genome of AM187. P value = $4.5 \times 10^{-6}$ by a two-tailed, unequal variance Student's t-test, N = 8. (**C**) *proA** copy number (left axis, solid lines) and growth rate (right axis, dotted lines) for population 3. Vertical dotted lines indicate when population genomic DNA was sequenced. Sequencing depth was 130x, 122x, 70x and 81x at the four points, respectively. The frequency of the *proA*** allele at each time point is noted above the plot.

The online version of this article includes the following figure supplement(s) for figure 5:

**Figure supplement 1.** Growth of the parental strain (AM187) and comparable strains with various *proA* alleles in the genome.

The fact that a mutation that improved the neo-ArgC activity of ProA* occurred in only one population was surprising considering that ProA* is the weak-link enzyme limiting growth rate. Because the growth rates of all eight populations improved substantially (*Figure 3*), mutations outside of the *proBA** operon must also be contributing to fitness.

## Some prevalent mutations in the evolved clones are not related to improved arginine synthesis

Population genome sequencing at the end of the experiment revealed that the final populations contained between 13 and 178 mutations at frequencies ≥ 5%, between 3 and 5 mutations at

**Table 2.** Kinetic parameters for GSA and NAGSA dehydrogenase activities of ProA, ProA*, and ProA**.

| | GSA activity (ProA) | | | NAGSA activity (neo-ArgC) | | |
|---|---|---|---|---|---|---|
| | $k_{cat}$ (s$^{-1}$) | $K_M$ (mM) | $k_{cat}/K_{M,GSA}$ (M$^{-1}$ s$^{-1}$) | $k_{cat}$ (s$^{-1}$) | $K_M$ (mM) | $k_{cat}/K_{M, NAGSA}$ (M$^{-1}$ s$^{-1}$) |
| WT | 16 ± 0.3 | 0.22 ± 0.01 | 72000 ± 2000 | 0.0083 ± 0.0009 | 0.30 ± 0.09 | 28 ± 9 |
| ProA* (E383A) | 0.0076 ± 0.0008 | 0.20 ± 0.04 | 37 ± 8 | 0.046 ± 0.002 | 0.076 ± 0.009 | 610 ± 74 |
| ProA** (E383A F372L) | 0.023 ± 0.005 | 0.42 ± 0.14 | 55 ± 22 | 0.21 ± 0.01 | 0.095 ± 0.011 | 2200 ± 260 |

[a] Values reported were calculated from a nonlinear least squares regression of three replicates at each substrate concentration ± standard error.

frequencies ≥ 30%, and between 1 and 4 fixed mutations (not including amplification of *proA*) (see *Figure 4—source data 1* for a list of mutations). We found several mutations in the same genes in different populations, suggesting that these mutations confer a fitness advantage.

The first mutation to appear in all populations was either an 82 bp deletion in the *rph* pseudo-gene directly upstream of *pyrE* or a C→T mutation in the intergenic region between *rph* and *pyrE*. These mutations occurred by 100 generations and prior to amplification of *proBA\**. PyrE is required for de novo synthesis of pyrimidine nucleotides (*Figure 2*). Both of these mutations have arisen in other *E. coli* evolution experiments, and have been shown to restore a known PyrE deficiency in the BW25113 *E. coli* strain (*Blank et al., 2014*; *Bonekamp et al., 1984*; *Conrad et al., 2009*; *Jensen, 1993*; *Knöppel et al., 2018*). The 82 bp deletion in *rph* increases growth rate of the parental AM187 strain by 55% (*Figure 4—figure supplement 2*). Thus, these mutations are general adaptations to growth in minimal medium and do not pertain to the selective pressures caused by the weak-link enzyme ProA*.

A mutation in *ygcB* occurred early in four populations. This mutation changes Ala390 to Val in Cas3, a nuclease/helicase in the Type I CRISPR/Cas system in *E. coli* (*Howard et al., 2011*). We introduced this mutation into the genome of the parent AM187 and compared the growth rates of the mutant and AM187 (*Figure 4—figure supplement 2*). Surprisingly, we saw no significant change in growth rate. Since this mutation appeared about the same time as the mutations upstream of *pyrE*, we wondered whether the *ygcB* mutation might only improve growth rate in the context of restored *pyrE* expression. Thus, we also tested the growth rate of a strain with the Cas3 mutation and the 82 bp deletion upstream of *pyrE*. Again, we saw no significant change in relative growth rate (*Figure 4—figure supplement 2*). Thus, the *ygcB* mutation is most likely a neutral hitchhiker. The most likely explanation for its prevalence is that it was present in a clade of the parental population that later rose to a high frequency when an additional beneficial mutation was acquired by one of its members.

## Mutations upstream of *argB* increase ArgB abundance

All eight final populations contained mutations in the intergenic region upstream of *argB* and downstream of *kan$^r$*. These mutations were fixed in two populations, and present at frequencies of 9–82% in the other populations (*Figure 6A*). ArgB (*N*-acetylglutamate kinase) catalyzes the second step in arginine synthesis, phosphorylation of *N*-acetylglutamate to form NAGP, the substrate for ArgC in wild-type *E. coli* and the substrate for ProA* in Δ*argC proA\* E. coli* (*Figure 2*).

We reintroduced six of the mutations upstream of *argB* into the parental strain AM187. The mutations increased growth rate by 36–61% (*Figure 6B*). Levels of mRNAs for *argB* and *argH*, which is immediately downstream of *argB*, were little affected by the mutations (*Figure 6C*). However, levels of ArgB protein increased 2.6–8.2-fold (*Figure 6D*). In contrast, ArgH levels increased only modestly. These data suggest that the mutations upstream of *argB* increase translational efficiency of *argB* mRNA. An increase in the amount of ArgB will increase production of NAGP, the substrate for the weak-link enzyme ProA* (*Figure 2*).

While increasing the level of *argB* is clearly beneficial in AM187, it is possible that replacing *argC* with the *kan$^r$* cassette might have altered expression of the downstream *argB*, artificially creating a situation in which ArgB activity is insufficient. Expression of *argB* and *argH* in AM187 is controlled by

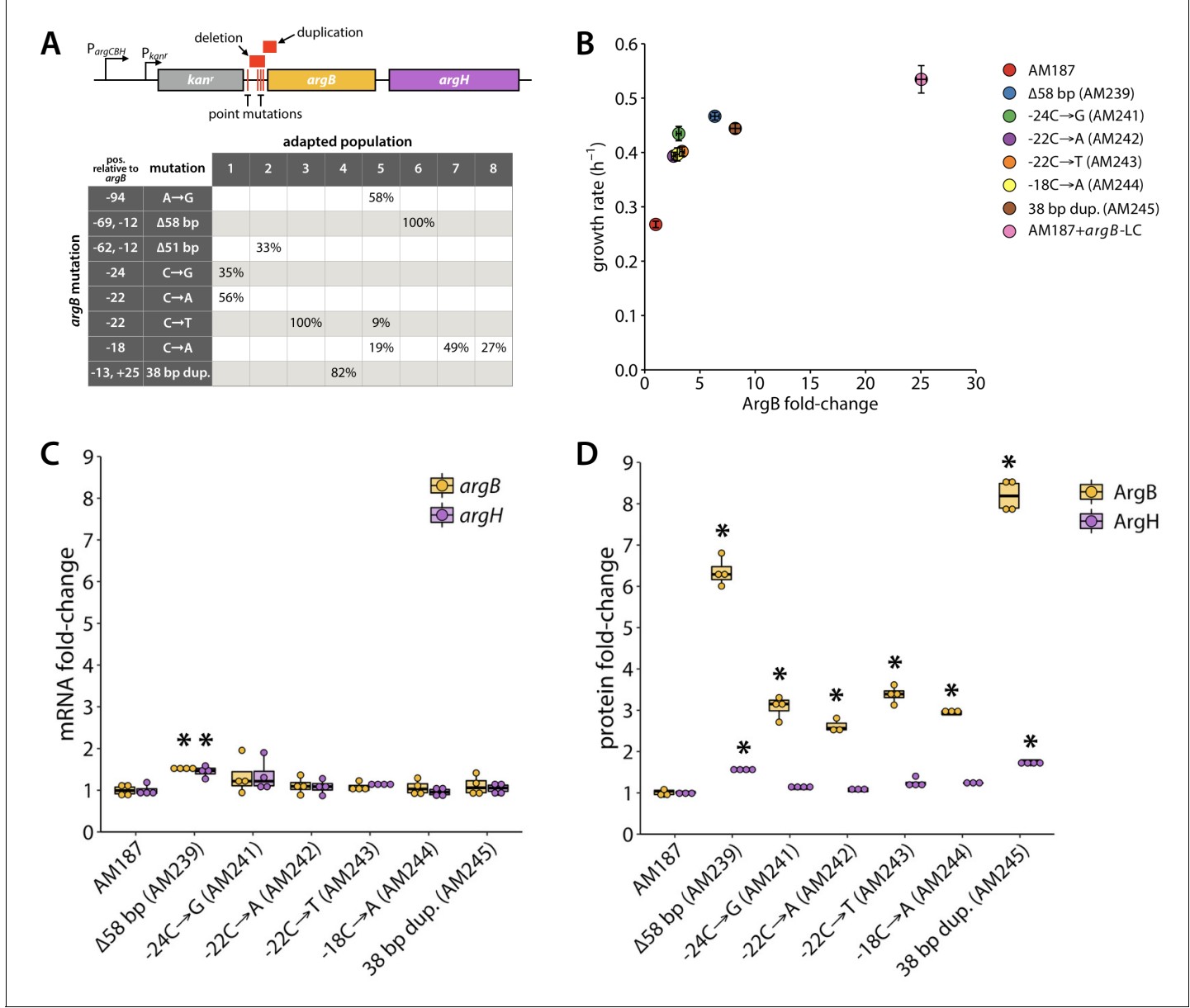

**Figure 6.** Several adaptive mutations occurred upstream of *argB*. (**A**) Locations of adaptive mutations (red) upstream of *argB* and *argH*. *argC* was replaced with *kan*^r in the parental strain AM187, giving this operon two promoters, one native to the operon ($P_{argCBH}$), and the other introduced with the *kan*^r gene ($P_{kanr}$). The table shows the percentages of each evolved population that contained a given *argB* mutation at the final time point. Six of the *argB* mutations were introduced into the genome of the parental AM187 strain and changes in growth rate (**B**), gene expression (**C**), and protein abundance (**D**) were determined (N = 4). Asterisks indicate values that were statistically different from those of the parental strain with *p* values ≤ 0.005 by a two-tailed, unequal variance Student's *t*-test. In (**B**), error bars represent ± SE. *argB*-LC denotes *argB* expression on a low-copy plasmid under control of *argB*'s native promoter.

The online version of this article includes the following figure supplement(s) for figure 6:

**Figure supplement 1.** The *argC::kan*^r construct in AM187 was replaced with a loss-of-function *argC* allele (top).

**Figure supplement 2.** *argB* mRNA secondary structure comparison.

**Figure supplement 3.** Calculated folding times of regions encompassing 30 nucleotides upstream of the *argB* start codon and 33 nucleotides of the *argB* coding region based on Kinfold simulations (*Wolfinger et al., 2004*).

**Figure supplement 4.** Difference in calculated energies for hybridization of sRNAs to mutant and parental *argB* sequences.

both their native promoter and a constitutive $kan^r$ promoter (**Figure 2—figure supplement 1**), possibly increasing transcription of the operon. Additionally, the different sequence of the intergenic region upstream of *argB* might influence translation of the *argB* mRNA. To determine the net effect of these two influences, we compared the levels of ArgB and ArgH in AM187 and a comparable strain (AM407) that lacks ArgC due to introduction of two stop codons in *argC* (**Figure 6—figure supplement 1**). The level of ArgH is 64% higher in AM187, probably due to increased transcription of the operon. In contrast, the level of ArgB is 2.3-fold lower, suggesting that the altered structure upstream of *argB* mRNA diminishes translation. Despite these changes, the growth rates of AM187 and AM407 are identical ($\mu = 0.27 \pm 0.01$ h$^{-1}$).

We further investigated the effect of altering ArgB levels on the growth rate of AM187 by expressing ArgB from a low-copy plasmid (**Figure 6B**). Growth rate of AM187 improves substantially when ArgB levels are increased by 25-fold, demonstrating that the beneficial effect of the mutations we observed in the evolved strains is not simply due to compensation for the 2.3-fold decrease in ArgB caused by replacement of *argC* with $kan^r$.

The increased translation efficiency of *argB* in the mutant strains might be due to decreased secondary structure around the Shine-Dalgarno site and start codon (**Bentele et al., 2013**; **Espah Borujeni et al., 2014**; **Goodman et al., 2013**). The *argB* mRNA, like 16% of γ-proteobacterial mRNAs (**Scharff et al., 2011**), lacks a canonical Shine-Dalgarno sequence, but the ribosome is expected to bind to a region encompassing the start codon and at least the upstream 8–10 nucleotides. We calculated the minimum free energy secondary structures of 140-nt RNA sequences encompassing the upstream intergenic region affected by the various mutations through 33 bp downstream of the *argB* start codon using CLC Main Workbench (**Figure 6—figure supplement 2**). Note that, although *argC* was replaced by $kan^r$ in the Keio strain used to construct AM187, the last 21 bp of *argC* and the 7 bp intergenic region between *argC* and *argB* are preserved. The FLP recognition target site downstream of $kan^r$ (used to remove the $kan^r$ cassette in the Keio strains [**Baba et al., 2006**]) forms a large stem-loop structure upstream of *argB*. However, this structure does not impact the region surrounding the putative *argB* ribosome binding site. The ribosome binding site is mostly sequestered in two stem-loops in the AM187 sequence. Four of the five point mutations occur in this region. The 58 bp and 51 bp deletions extend into this region, and the 38 bp duplication begins 13 bp upstream of the *argB* start codon within this region. For five of the eight mutant structures, the probability that the 5'-UTR upstream of the start codon is sequestered in the lowest free-energy structure is decreased relative to the parental sequence (**Figure 6—figure supplement 2**); the increased accessibility of this region should increase translation efficiency. However, for three mutants ($-94$ A$\rightarrow$G, $-22$ C$\rightarrow$A, and $-18$ C$\rightarrow$A), this region is equally or more likely to be sequestered in a stem-loop. The thermodynamic stability of this region is clearly not the only factor responsible for the effects of the mutations upstream of *argB*.

We also considered the possibility that mutations upstream of *argB* might increase expression by increasing ribosome drafting (binding of a ribosome to the unfolded mRNA emerging behind a preceding ribosome before the mRNA folds and obscures the Shine-Dalgarno sequence) (**Espah Borujeni and Salis, 2016**). **Figure 6—figure supplement 3** shows the predicted folding times of 63 nt RNA sequences centered around the start codon for each mutant except the $-94$ A$\rightarrow$G mutant. (The point mutation at $-94$ relative to the start codon is outside of the window used for the calculation.) The significantly slower folding of three of the mutant RNAs (51 bp deletion, $-24$ C$\rightarrow$G, and $-18$ C$\rightarrow$A) should increase translation efficiency. For two of the mutants for which folding rate is either the same (the 58 bp deletion) or increased (the 38 bp duplication), the secondary structure prediction shown in **Figure 6—figure supplement 2** suggests that the ribosome binding site is less likely to be sequestered in a hairpin. Thus, the effects of 6 of the eight mutations can be explained by a decrease in secondary structure stability around the ribosome binding site, a decrease in the folding rate of the mRNA in this region, or both. The effects of the $-94$ A$\rightarrow$G and $-22$ C$\rightarrow$A mutations, however, cannot be explained by either mechanism.

A final possibility is that translation efficiency could be increased if a mutation weakens an sRNA: mRNA interaction that blocks the ribosome binding site. There is no known physiological interaction between an sRNA and the *argB* mRNA, so this explanation is unlikely. Alternatively, a mutation might strengthen a sRNA:mRNA interaction that competes with a mRNA secondary structure that inhibits ribosome binding, thereby increasing the accessibility of the ribosome binding site. We explored the effects of the mutations upstream of *argB* on the predicted binding energies of 65

annotated sRNAs to the RNA sequences used for the secondary structure predictions (*Figure 6—figure supplement 4*) using the IntaRNA algorithm (*Busch et al., 2008*; *Mann et al., 2017*; *Raden et al., 2018*; *Wright et al., 2014*). The calculated binding energy sums the energy needed to denature sRNA and mRNA secondary structures and the hybridization energy of the unfolded sRNA and mRNA. None of the 65 sRNAs had a calculated binding energy for the parental *argB* region in the range of those for known physiological interactions between sRNAs and target mRNAs (e.g. $-16.1$ kcal/mol, ChiX and *dpiB*; $-13.0$ kcal/mol, OmrA and *csgD*; $-14.9$ kcal/mol, DsrA and *rpoS*), $-14.3$ kcal/mol, RprA and *rpoS*), with the strongest binding energy being $-7.4$ kcal/mol. Mutations decreased the predicted binding energy to $<-11$ kcal/mol for only one sRNA, RyfA, and only for the 58 bp deletion, 38 bp duplication and $-94$ A$\rightarrow$G point mutation. Binding of RyfA was predicted to increase in a region that is not involved in the secondary structure around the ribosome binding site (*Figure 6—figure supplement 4B*). Thus, differences in binding to sRNAs are unlikely to be responsible for the changes in translation efficiency.

## Mutations in *carB* either increase activity or impact allosteric regulation

We found eight different mutations in *carB* in six of the evolved populations: four missense mutations, three deletions ($\geq$12 bp), and one 21 bp duplication (*Figure 7A*). CarB, the large subunit of carbamoyl phosphate synthetase (CPS), forms a complex with CarA to catalyze production of carbamoyl phosphate from glutamine, bicarbonate, and two molecules of ATP (*Equation 1*).

$$\mathrm{Gln + HCO_3^- + 2\,ATP \longrightarrow carbamoyl\,phosphate + Glu + 2\,ADP + P_i} \tag{1}$$

Synthesis of carbamoyl phosphate involves four reactions that take place in three separate active sites connected by a molecular tunnel of ~100 Å in length (*Thoden et al., 2002*). CarA catalyzes hydrolysis of glutamine to glutamate and ammonia (*Equation 2*). CarB phosphorylates bicarbonate to form carboxyphosphate in its first active site (*Equation 3*). Ammonia from the CarA active site is channeled to CarB, where it reacts with carboxyphosphate to form carbamate (*Equation 4*). Carbamate migrates to a second active site within CarB, where it reacts with ATP to form carbamoyl phosphate and ADP (*Equation 5*).

$$\mathrm{Gln + H_2O \longrightarrow NH_3 + Glu} \tag{2}$$

$$\mathrm{HCO_3^- + ATP \longrightarrow carboxyphosphate + ADP} \tag{3}$$

$$\mathrm{NH_3 + carboxyphospate \longrightarrow carbamate + P_i} \tag{4}$$

$$\mathrm{carbamate + ATP \longrightarrow carbamoyl\,phosphate + ADP} \tag{5}$$

Carbamoyl phosphate feeds into both the pyrimidine and arginine synthesis pathways and its production is regulated in response to an intermediate or product of both pathways (*Figure 2*, *Figure 7B*), as well as by IMP (*Pierrat and Raushel, 2002*). CarB is inhibited by UMP (a pyrimidine) and moderately activated by IMP (a purine). UMP and IMP compete to bind the same region of CarB (*Eroglu and Powers-Lee, 2002*). The net effect is inhibition of CarB when pyrimidine levels are high and activation when purine levels are high. The allosteric effects of UMP and IMP are dominated, however, by activation by ornithine. Ornithine, an intermediate in arginine synthesis that reacts with carbamoyl phosphate, binds to and activates CarB even when UMP is bound (*Figure 7C*) (*Braxton et al., 1999*; *Eroglu and Powers-Lee, 2002*). Thus, flux into arginine synthesis can be maintained even when pyrimidine levels are sufficient.

Seven of the eight mutations found in *carB* affect residues in the allosteric domain of CarB. The other mutation changes Gly369, which is immediately adjacent to the allosteric region, to Val (*Figure 7C*).

The kinetic parameters for carbamoyl phosphate synthetase (CPS) activity (determined as the glutamine- and bicarbonate-dependent ATPase activity [*Equation 1*]) of all eight CPS variants are shown in *Table 3*. All mutations decreased $k_{cat}/K_{m,ATP}$ by 34–63%, with the exception of the mutation that changes Lys966 to Glu, which nearly doubles $k_{cat}/K_{m,ATP}$. None of the mutations affected

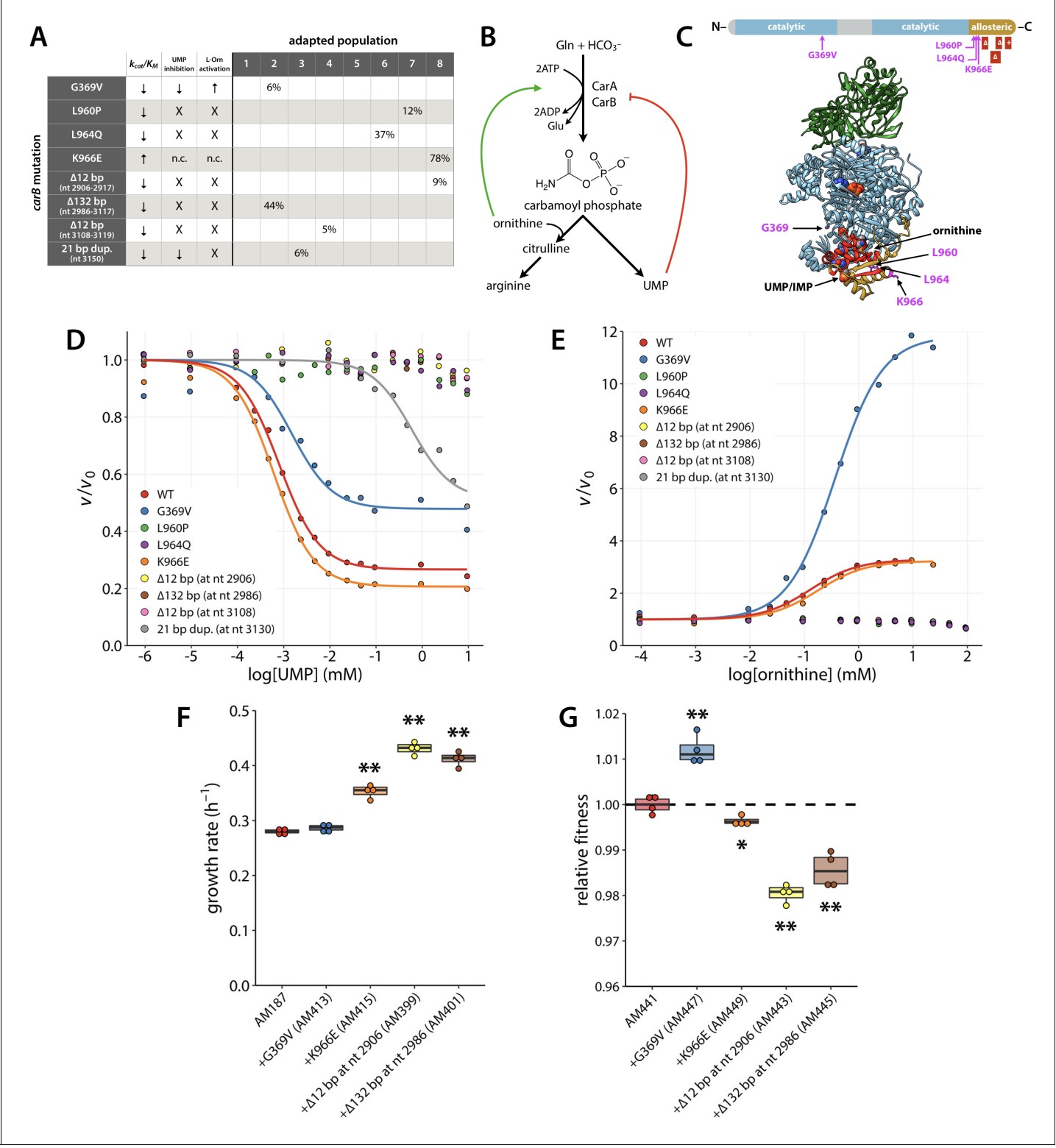

**Figure 7.** Several adaptive mutations occurred in *carB*. (**A**) Maximum percentage of each *carB* mutation found in the population at any time during the evolution. Nucleotide (nt) numbers below mutant descriptions indicate where deletions or duplications occurred in the 3222 nt *carB*. Arrows indicate that a kinetic parameter was either increased or decreased in the variant enzyme relative to the wild-type enzyme. X, loss of activity; n.c., no change. (**B**) Allosteric regulation of carbamoyl phosphate synthetase. CarA and CarB are the small and large subunits of carbamoyl phosphate synthetase, respectively. (**C**) CarB functional domains (top) and crystal structure of *E. coli* CarAB (PDB 1CE8, bottom) (*Thoden et al., 1999*). Green, CarA; blue,

*Figure 7 continued on next page*

*Figure 7 continued*

CarB; gold, allosteric domain of CarB; red, residues that are deleted or duplicated in the adapted strains; magenta, point mutations that occur in the adapted strains. IMP and ornithine bound to the allosteric domain are shown as spheres. One of the two bound ATP molecules can be seen as spheres in the center of CarB. (D–E) Influence of UMP and L-ornithine on the ATPase activity of CarAB. $v_0$; reaction rate in the absence of ligand. Each point represents the average of three technical replicates. (F) Growth rates of the parental AM187 strain and strains in which *carB* mutations had been introduced into the genome of AM187. (G) Relative fitness of AM441 (*E. coli* BW25113 containing the $\Delta$82 bp mutation upstream of *pyrE*) and strains in which the *carB* mutations had been introduced into the genome of AM441. Asterisks in (F) and (G) indicate differences with p values < 0.03 (single asterisk) or $\leq$ 0.001 (double asterisk) by a two-tailed, unequal variance Student's *t*-test, N = 4.

The online version of this article includes the following figure supplement(s) for figure 7:

**Figure supplement 1.** Ratio of ADP and carbamoyl phosphate (CP) produced by wild-type and mutant carbamoyl phosphate synthetases.
**Figure supplement 2.** Effects of arginine (5.2 mM) and uracil (0.2 mM) on growth rates of AM187, AM327 and AM441.

the enzyme's ability to couple ATP hydrolysis with carbamoyl phosphate production (*Figure 7—figure supplement 1*).

We measured the effect of mutations on UMP inhibition and ornithine activation of CPS (*Table 3*, *Figure 7D–E*). Regulation of the K966E variant, the enzyme for which $k_{cat}/K_{m,ATP}$ was nearly doubled, was minimally affected. Five of the variants showed complete loss of allosteric regulation. The variant with the 21 bp duplication retained modest inhibition by UMP, but only at very high concentrations of UMP; the apparent $K_{d,UMP}$ was increased by 740-fold. Similarly, G369V CPS retained partial inhibition by UMP. While the apparent $K_{d,UMP}$ of the G369V enzyme only doubled, this variant showed a 3.5-fold increase in activation at high ornithine concentrations.

The eight *carB* mutations result in increased CPS activity via three different mechanisms: (1) increased catalytic turnover (K966E); (2) increased activation by ornithine (G369V); and (3) decreased inhibition by UMP (L960P, L964Q, $\Delta$12 bp at nt 2906, $\Delta$132 bp at nt 2986, $\Delta$12 bp at nt 3108, and 21 bp duplication at nt 3130). In vivo, the increased CPS activity would be expected to increase the level of carbamoyl phosphate, and thereby increase the rate at which ornithine transcarbamoylase produces citrulline from carbamoyl phosphate and ornithine downstream of the ProA* bottleneck in the arginine synthesis pathway (*Figure 2*).

We introduced four of the *carB* mutations into the parental strain AM187 to confirm that they were beneficial. Three of the mutations (K966E, $\Delta$12 bp at nt 2906, and $\Delta$132 bp at nt 2986) increased growth rate (*Figure 7F*). The two mutations that caused loss of UMP inhibition ($\Delta$12 bp at nt 2906 and $\Delta$132 bp at nt 2986) showed the greatest increase in growth rate (47–54%). The mutation that increased CPS catalytic activity (K966E) increased growth rate by 26%.

**Table 3.** Kinetic parameters[a] for the glutamine- and bicarbonate-dependent ATPase reaction of wild-type and variant carbamoyl phosphate synthetases.

| Enzyme | $K_{M,\ ATP}$ (mM) | $k_{cat}$ ($s^{-1}$) | $k_{cat}/K_{M,ATP}$ ($M^{-1}\ s^{-1}$) | UMP $K_d$ ($\mu$M) | UMP $a$ | ornithine $K_d$ ($\mu$M) | ornithine $a$ |
|---|---|---|---|---|---|---|---|
| WT | 1.05 ± 0.08 | 13.5 ± 0.3 | 12.9 (±1.0)×$10^3$ | 0.81 ± 0.04 | 0.27 ± 0.01 | 130 ± 7 | 3.28 ± 0.03 |
| G369V | 3.31 ± 0.25 | 21.5 ± 0.7 | 6.51 (±0.54)×$10^3$ | 1.53 ± 0.29 | 0.48 ± 0.02 | 372 ± 20 | 11.8 ± 0.13 |
| L960P | 1.12 ± 0.05 | 9.10 ± 0.15 | 8.12 (±0.41)×$10^3$ | *na* | *na* | *na* | *na* |
| L964Q | 1.25 ± 0.08 | 8.04 ± 0.17 | 6.41 (±0.42)×$10^3$ | *na* | *na* | *na* | *na* |
| K966E | 0.97 ± 0.06 | 20.6 ± 0.4 | 21.2 (±1.4)×$10^3$ | 0.61 ± 0.04 | 0.21 ± 0.01 | 181 ± 34 | 3.23 ± 0.08 |
| $\Delta$12 bp (at nt 2906)[b] | 1.09 ± 0.06 | 4.80 ± 0.09 | 4.39 (±0.25)×$10^3$ | *na* | *na* | *na* | *na* |
| $\Delta$132 bp (at nt 2986) | 1.16 ± 0.06 | 6.47 ± 0.11 | 5.57 (±0.31)×$10^3$ | *na* | *na* | *na* | *na* |
| $\Delta$12 bp (at nt 3108) | 1.30 ± 0.10 | 5.86 ± 0.16 | 4.51 (±0.37)×$10^3$ | *na* | *na* | *na* | *na* |
| 21 bp dup. (at nt 3130) | 1.40 ± 0.12 | 9.70 ± 0.30 | 6.94 (±0.64)×$10^3$ | 597 ± 133 | 0.51 ± 0.03 | *na* | *na* |

[a] Values reported ± standard error. Values for $K_d$ and $a$ for UMP and ornithine were determined by fitting the data to the following equation: $v/v_0 = (aL + K_d)/(L + K_d)$, where L is the ligand concentration, $v$ is the initial reaction rate, $v_0$ is the initial reaction rate in the absence of ligand, $a$ is $v/v_0$ at infinite L, and $K_d$ is the apparent dissociation constant. No $K_d$ or $a$ values are given (indicated by *na*) when inhibition by the allosteric ligand was too weak to measure.
[b] Nucleotide (nt) numbers refer to the position of deletions or duplications in *carB*.

The G369V mutation does not improve growth rate of AM187, which is not surprising because its major effect is to increase ornithine activation at high ornithine concentrations. In AM187, the ornithine concentration is likely to be low due to the bottleneck in the arginine synthesis pathway caused by ProA*. Thus, increasing ornithine activation of CPS would have little effect. We suspect that this mutation may only be beneficial after gene amplification increases ProA* levels.

We also considered the possibility that the *carB* mutations are beneficial because they boost production of carbamoyl phosphate for pyrimidine synthesis. *E. coli* K strains are known to have a pyrimidine synthesis deficiency due to a mutation in *rph* that impacts transcription of the downstream *pyrE*. The *rph-pyrE* mutation that occurred first in all populations is known to correct the pyrimidine synthesis deficiency (*Blank et al., 2014*; *Bonekamp et al., 1984*; *Conrad et al., 2009*; *Jensen, 1993*; *Knöppel et al., 2018*). However, pyrimidine synthesis might still be compromised in our evolving strains because levels of ornithine, the most important allosteric activator of CPS, are low due to the inefficiency of ProA*. To determine whether the growth defect of the AM187 strain with the Δ82 bp *rph-pyrE* mutation is due to limited synthesis of pyrimidines, arginine, or both, we tested the effect of adding uracil, arginine, or both on growth of the parental AM187 strain and AM327 (the AM187 strain with the Δ82 bp *rph-pyrE* mutation) (*Figure 7—figure supplement 2*). AM327 grows 60% faster than AM187, presumably due to improved pyrimidine synthesis. Adding uracil to the medium increased the growth rate of AM187, but did not affect the growth rate of AM327, suggesting that pyrimidine synthesis is no longer insufficient after acquisition of the Δ82 bp *rph-pyrE* mutation. In contrast, adding arginine restored growth of both strains to wild-type levels. These results suggest that at the time the *carB* mutations occurred, they improved arginine synthesis rather than pyrimidine synthesis.

Mutations that impact the elaborate allosteric regulation of CarB would be expected to be detrimental after arginine synthesis is restored. To test this hypothesis, we introduced four of the *carB* mutations into the genome of AM441, a wild-type strain into which the *rph-pyrE* mutation had been introduced, and measured their effects on fitness using a competitive fitness assay (*Figure 7G*). The two deletion mutations that abolished allosteric regulation (Δ12 bp at nt 2906 and Δ132 bp at nt 2986) significantly decreased growth rate. The K966E mutation, which increases $k_{cat}/K_{M,ATP}$ by 64% but shifts the balance between the regulatory effects of UMP and ornithine by modestly increasing UMP inhibition and decreasing ornithine activation, also slightly decreases growth rate. The G369V mutation, which diminishes inhibition by UMP but substantially increases activation by ornithine, actually increased growth rate, suggesting that the balance between the regulatory effects of UMP and ornithine in the wild-type CarB may not be optimal after the *rph-pyrE* mutation improves pyrimidine synthesis. These results suggest that many of the *carB* mutations provide a fitness improvement when arginine synthesis is compromised, but will be detrimental once an efficient neo-ArgC has emerged.

## Discussion

Recruitment of promiscuous enzymes to serve new functions followed by mutations that improve the promiscuous activity has been a dominant force in the diversification of metabolic networks (*Copley, 2017*; *Glasner et al., 2006*; *Khersonsky and Tawfik, 2010*; *O'Brien and Herschlag, 1999*; *Rauwerdink et al., 2016*). New enzymes may be important for fitness or even survival when an organism is exposed to a novel toxin or source of carbon or energy, or when synthesis of a novel natural product enables manipulation of competing organisms or the environment. This process also contributes to non-orthologous gene replacement, which can occur when a gene is lost during a time in which it is not required, but its function later becomes important again and is replaced by recruitment of a non-orthologous promiscuous enzyme (*Albalat and Cañestro, 2016*; *Ferla et al., 2017*; *Juárez-Vázquez et al., 2017*; *Newton et al., 2018*; *Olson, 1999*).

We have modeled a situation in which a new enzyme is required by deleting *argC*, which is essential for synthesis of arginine in *E. coli*. Previous work showed that a promiscuous activity of ProA is the most readily available source of neo-ArgC activity that enables Δ*argC E. coli* to grow on glucose as a sole carbon source. However, a point mutation that changes Glu383 to Ala is required to elevate the promiscuous activity to a physiologically useful level. This mutation substantially damages the native function of the enzyme, creating an inefficient bifunctional enzyme whose poor catalytic abilities limit growth rate on glucose. It is important to note that the decrease in the efficiency of the

native reaction may be a critical factor in the recruitment of ProA because it will diminish inhibition of the newly important reaction by the native substrate (*Khanal et al., 2015*; *McLoughlin and Copley, 2008*).

We chose to carry out evolution of a Δ*argC proA\** strain with a previously identified promoter mutation upstream of *proA\** in glucose in the presence of proline to specifically address the evolution of an efficient neo-ArgC. After 470–1000 generations of evolution, growth rate was increased by ~3 fold in all eight replicate cultures. We have focused on five types of genetic changes that clearly increase fitness (*Figure 8*): (1) mutations upstream of *pyrE*; (2) amplification of a variable region of the genome surrounding the *proBA\** operon; (3) a mutation in *proA\** that changes Phe372 to Leu; (4) mutations upstream of *argB*; and (5) mutations in *carB*. (Each of the final populations contains additional mutations that may also contribute to fitness, but these mutations were typically found in low abundance and/or in only one population.) The mutations upstream of *pyrE* occurred first (within 100 generations) and have previously been shown to be a general adaptation of *E. coli* BW25113 to growth in minimal medium (*Blank et al., 2014*; *Conrad et al., 2009*; *Jensen, 1993*; *Knöppel et al., 2018*). The other four types of mutations are specific adaptations to the bottleneck in arginine synthesis caused by substitution of the weak-link enzyme ProA\* for ArgC in this strain. Interestingly, only two of these—gene amplification and the mutation in *proA\**—directly involve the weak-link enzyme ProA\*.

Surprisingly, we saw evolution of *proA\** towards a more efficient neo-ArgC in only one population (*Figure 5*). In this population, *proA* copy number dropped from ~7 to ~3 within 100 generations. This pattern is consistent with the IAD model; copy number is expected to decrease as mutations increase the efficiency of the weak-link activity. However, the fact that copy number did not return to one implies that the neo-ArgC activity of ProA\*\* is not sufficient to justify a single copy of the gene.

Because ~3 copies of *proA\*\** remained in the population and the progenitor *proA\** was not detectable (*Figure 5C*), all copies in the amplified array have clearly acquired the mutation that changes Phe372 to Leu – that is, the more beneficial *proA\*\** allele has 'swept' the amplified array. This observation has important implications for the IAD model. In the original conception of the IAD model, it was proposed that amplification of a gene increases the opportunity for different beneficial mutations to occur in different copies, and then for recombination to shuffle these mutations (*Bergthorsson et al., 2007*; *Francino, 2005*). These phenomena would increase the rate at which sequence space can be searched and thereby the rate at which a new enzyme evolves. In order for this to occur, however, it would be necessary for individual alleles to acquire different beneficial mutations before recombination occurred. This scenario is inconsistent with the relative frequencies of point mutations and recombination between large homologous regions in an amplified array (*Anderson and Roth, 1981*; *Reams et al., 2010*). Point mutations occur at a frequency between $10^{-9}$ and $10^{-10}$ per nucleotide per cell division depending on the genomic location (*Jee et al., 2016*), and thus between $10^{-6}$ and $10^{-7}$ per gene per cell division for a gene the size of *proA*. If 10 copies of an evolving gene were present, then the frequency of mutation in a single allele would be between $10^{-5}$ and $10^{-6}$ per cell division. Homologous recombination after an initial duplication event is orders of magnitude more frequent, occurring in ~1 of every 100 cell divisions (*Reams et al., 2010*). Thus, homologous recombination between replicating chromosomes in a cell could result in a selective allelic sweep (*Figure 9*) long before a second beneficial mutation occurs in a different allele in the amplified array. This is indeed the result that we observed; heterozygosity among *proA\** alleles was lost within 500 generations (*Figure 5C*). More recent papers depict selective amplification of beneficial alleles before acquisition of additional mutations (*Andersson et al., 2015*; *Näsvall et al., 2012*); our results support this revision of the original IAD model. It is possible that alleles encoding enzymes that are diverging toward two specialists might recombine to explore combinations of mutations. However, such recombination might not accelerate evolution of a new enzyme, as mutations that lead toward one specialist enzyme would likely be incompatible with those that lead toward the other specialist enzyme.

While growth rate improved substantially in all populations, a beneficial mutation in *proA\** arose in only one, suggesting either that mutations that improve the neo-ArgC activity are uncommon, or that their fitness effects are smaller than those caused by mutations elsewhere in the genome that also improve arginine synthesis. We identified two primary mechanisms that apparently improve arginine synthesis without affecting the efficiency of the weak-link enzyme ProA\* itself.

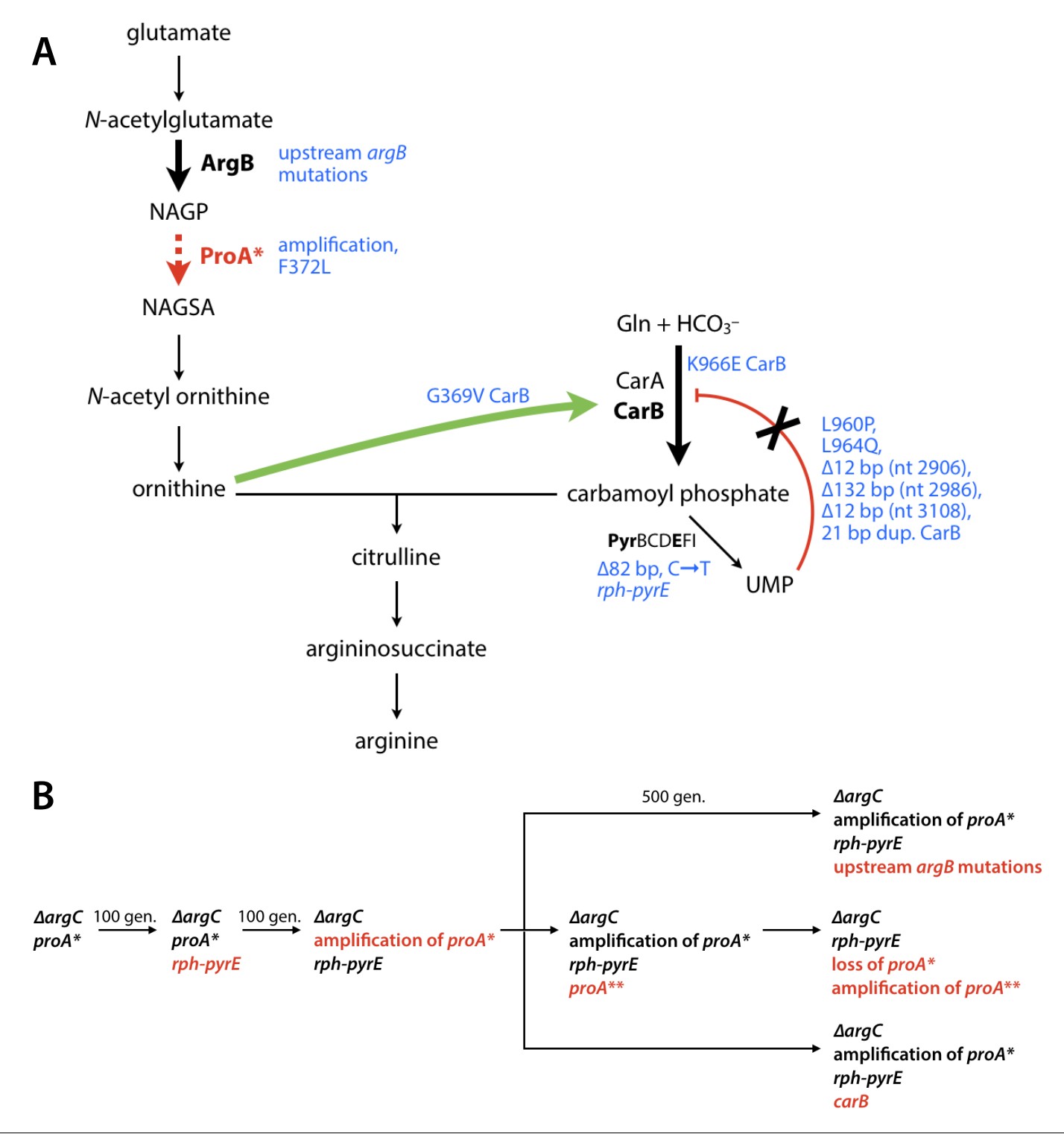

**Figure 8.** Adaptive mutations are predicted to increase flux through the arginine synthesis pathway. (**A**) The pathway bottleneck enzyme ProA* is shown in red. Steps in the arginine synthesis pathway that are affected by adaptive mutations (blue) are highlighted by bold arrows. (**B**) Trajectories for adaptive mutations observed during 470–1000 generations of evolution. Red text denotes a new mutation.

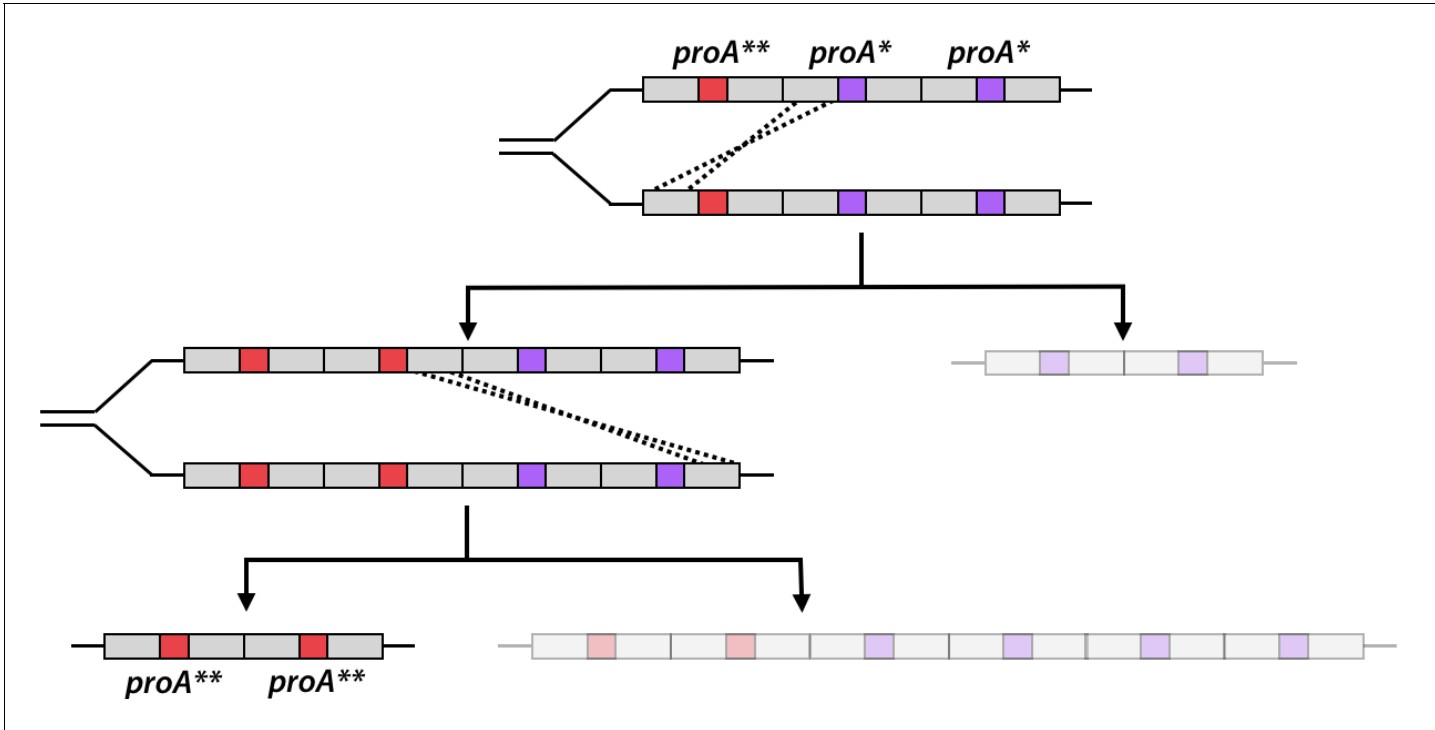

**Figure 9.** Homologous recombination of an amplified *proA** array with one *proA*** allele can rapidly lead to a daughter cell with only *proA*** alleles. Each arrow represents one homologous replication event. The genotype of the less-fit daughter cell from each recombination event is grayed out.

We identified eight mutations upstream of *argB*; the six we tested improved growth rate by 36–61% and increased the abundance of ArgB by 2.6–8.2-fold. Notably, ArgB levels were increased even though the levels of *argB* mRNA were unchanged (*Figure 6*). The increase in protein levels without a concomitant increase in mRNA levels suggests that these mutations impact the efficiency of translation. Secondary structure around the translation initiation site plays a key role because this region must be unfolded in order to bind to the small subunit of the ribosome (*Hall et al., 1982*; *Scharff et al., 2011*). Indeed, a study of the predicted secondary structures of 5000 genes from bacteria, mitochondria and plastids, many of which lack canonical Shine-Dalgarno sequences (as does *argB*), showed that secondary structure around the start codon is markedly less stable than up- or down-stream regions (*Bentele et al., 2013*; *Espah Borujeni et al., 2014*; *Goodman et al., 2013*; *Scharff et al., 2011*). Our computational studies of the effect of mutations on the predicted lowest free energy secondary structures of the region surrounding the start codon of *argB* suggest that the thermodynamic stability of this region plays a role in the beneficial effects of most of the observed mutations (*Figure 6—figure supplement 2*). In addition, three of the mutations slow the predicted rate of mRNA folding around the start codon, which would increase the probability of ribosomal drafting (*Figure 6—figure supplement 3*). Both effects would lead to an increase in ArgB abundance, which should increase the concentration of the substrate for the weak-link ProA*, thereby pushing material through this bottleneck in the arginine synthesis pathway.

The adaptive mutations in *carB* increase catalytic turnover, decrease inhibition by UMP, or increase activation by ornithine of CPS. All of these effects should increase the level of CPS activity in the cell and consequently the level of carbamoyl phosphate. Why would this be advantageous? Ornithine transcarbamoylase catalyzes formation of citrulline from carbamoyl phosphate and ornithine, which will be in short supply due to the upstream ProA* bottleneck (*Legrain and Stalon, 1976*). If ornithine transcarbamoylase is not saturated with respect to carbamoyl phosphate, then increasing carbamoyl phosphate levels should increase citrulline production and thereby increase flux into the lower part of the arginine synthesis pathway. Although we do not know the concentration of carbamoyl phosphate in vivo, and thus cannot determine whether ornithine transcarbamoylase is saturated (the $K_M$ for carbamoyl phosphate is 360 µM *Baur et al., 1990*), the occurrence of so

many mutations that increase CPS activity and growth rate supports the notion that they lead to an increase in carbamoyl phosphate that potentiates flux through the arginine synthesis pathway.

The majority of adaptive mutations we observed in *carB* cause loss of the exquisite allosteric regulation that controls flux through this important step in pyrimidine and arginine synthesis. This tight regulation likely evolved due to the energetically costly reaction catalyzed by CPS, which consumes two ATP molecules (*Figure 7B*). While a constitutively active CPS is beneficial in the short term to improve arginine synthesis, it is detrimental once arginine production no longer limits growth. When we introduced four of the *carB* mutations into the genome of strain AM441 (wild-type *E. coli* containing the *rph-pyrE* mutation), three of the four mutations (K966E, Δ12 bp at nt 2906, and Δ132 bp at nt 2986) decreased fitness (*Figure 7G*). Notably, the mutations that impaired growth rate in the wild-type background were the same mutations that increased fitness in AM187. We term mutations such as these 'expedient' mutations because they provide a quick fix when cells are under strong selective pressure, but at a cost to a previously well-evolved function. The damage caused by expedient mutations may be repairable later by reversion, compensatory mutations or horizontal gene transfer. Interestingly, the latter two repair processes may contribute to sequence divergence between organisms that has typically been attributed to neutral drift, but rather may be due to scars left from previous selective processes.

A particularly striking conclusion from this work is that most of the mutations that improved fitness under these selective conditions did not impact the gene encoding the weak-link enzyme, but rather compensated for the bottleneck in metabolism by other mechanisms. The prevalence of adaptive mutations outside of *proA\** is likely a result of both a limited number of adaptive routes for improving the neo-ArgC activity of ProA\* and a larger target size for other beneficial mutations (*Ilhan et al., 2019*). Although the single mutation that we observed in *proA\** is highly beneficial, the paucity of *proA\** mutations suggests that only a small number of mutations at specific positions may improve the enzyme's activity. Directed evolution experiments often show a limited number of paths for improvement of enzymatic activity (*Aharoni et al., 2005*; *Sunden et al., 2015*; *Weinreich et al., 2006*), which reflects the stringent requirements for optimal placement of substrate-binding and catalytic residues in active sites. In contrast, there are multiple ways in which allosteric inhibition of CarB by UMP can be lost, and multiple ways in which translation efficiency of *argB* mRNA can be improved.

Not surprisingly, the process of evolution of a new enzyme by gene duplication and divergence does not take place in isolation, but is inextricably intertwined with mutations in the rest of the genome. The ultimate winner in a microbial population exposed to a novel selective pressure that requires evolution of a new enzyme may be the clone that succeeds in evolving an efficient enzyme while accumulating the least damaging, or at least the most easily repaired, expedient mutations.

## Materials and methods

### Materials

Common chemicals were purchased from Sigma-Aldrich (St. Louis, MO) and Fisher Scientific (Fair Lawn, NJ).

NAGSA was synthesized enzymatically from *N*-acetylornithine using *N*-acetylornithine aminotransferase (ArgD) in a 25 mL reaction as described previously by *Khanal et al. (2015)* and stored at −70°C. NAGSA concentrations were determined using the *o*-aminobenzaldehyde assay as described previously (*Albrecht et al., 1962*; *Mezl and Knox, 1976*).

GSA was synthesized enzymatically from L-ornithine using *N*-acetylornithine aminotransferase (ArgD) as described previously by *Khanal et al. (2015)* and stored at −70°C. GSA concentrations were determined using the *o*-aminobenzaldehyde assay as described previously (*Albrecht et al., 1962*; *Mezl and Knox, 1976*).

Plasmids and primers used in this study are listed in *Supplementary file 1* and *Supplementary file 2*.

### Strains and culture conditions

Strains used in this study are listed in *Table 1*. *E. coli* cultures were routinely grown in LB medium at 37°C with 20 µg/mL kanamycin, 100 µg/mL ampicillin, 20 µg/mL chloramphenicol, or 10 µg/mL

tetracycline, as required. Evolution of strain AM187 was performed at 37°C in M9 minimal medium containing 0.2% glucose, 0.4 mM proline, and 20 µg/mL kanamycin (Evolution Medium).

## Strain construction

The parental strain for the evolution experiment (AM187) was constructed from the Keio collection argC::kan^r E. coli BW25113 strain (**Baba et al., 2006**). The fimAICDFGH and csgBAC operons were deleted (to slow biofilm formation), and the M2 proBA promoter mutation (**Kershner et al., 2016**) and the point mutation in proA that changes Glu383 to Ala (**McLoughlin and Copley, 2008**) were inserted into the genome using the scarless genome editing technique described in **Kim et al. (2014)**. We initially hoped to measure proA* copy number during adaptation using fluorescence, although ultimately qPCR proved to be a better approach. Thus, we inserted yfp downstream of proA* under control of the P3 promoter (**Mutalik et al., 2013**) and with a synthetically designed ribosome binding site (**Espah Borujeni et al., 2014**; **Salis et al., 2009**). A double transcription terminator (BioBrick Part: BBa_B0015) was inserted immediately downstream of proBA* to prevent read-through transcription of yfp (**Figure 2—figure supplement 1**). We also inserted a NotI cut site immediately downstream of proA* to enable cloning of individual proA* alleles after amplification if necessary. A Fis binding site located 32 bp downstream of proA was preserved because it might impact proA transcription. The NotI-2xTerm-yfp cassette was inserted downstream of proA* using the scarless genome editing technique described in **Kim et al. (2014)**. The genome of the resulting strain AM187 was sequenced to confirm that there were no unintended mutations and deposited to NCBI GenBank under accession number CP037857.

Strain AM209 was constructed from E. coli BL21(DE3) for expression of wild-type and mutant ProAs. We deleted argC and proA to ensure that any activity measured during in vitro assays was not due to trace amounts of ArgC or wild-type ProA. To accomplish these deletions, we amplified and gel-purified DNA fragments containing antibiotic resistance genes (kanamycin and chloramphenicol for deletion of argC and proA, respectively) flanked by 200–400 bp of sequences homologous to the upstream and downstream regions of either argC or proA. E. coli BL21(DE3) cells containing pSIM27 (**Datta et al., 2006**) – a vector containing heat-inducible λ Red recombinase genes – were grown in LB/tetracycline at 30°C to an OD of 0.2–0.4 and then incubated in a 42°C shaking water bath for 15 min to induce expression of λ Red recombinase genes. The cells were then immediately subjected to electroporation with 100 ng of the appropriate linear DNA mutation cassette. Successful transformants were selected on either LB/kanamycin or LB/chloramphenicol plates.

Strain AM267 was constructed by deleting carAB from E. coli BL21 for expression of wild-type and mutant carbamoyl phosphate synthetases (CPS) to ensure that any activity measured during in vitro assays was not due to trace amounts of wild-type CPS. To accomplish the deletion, we amplified and gel-purified a DNA fragment containing the kanamycin resistance gene flanked by 40 bp of sequence homologous to the upstream and downstream regions of carAB. E. coli BL21 cells containing pSIM5 (**Datta et al., 2006**) – a vector carrying heat-inducible λ Red recombinase genes – were grown in LB/chloramphenicol at 30°C to an OD of 0.2–0.4 and then incubated in a 42°C shaking water bath for 15 min to induce expression of λ Red recombinase genes. The cells were then immediately subjected to electroporation with 100 ng of the appropriate linear DNA mutation cassette. Successful transformants were selected on LB/kanamycin plates.

Most mutations observed during the evolution experiment were introduced into the parental AM187 strain using the scarless genome editing protocol described in **Kim et al. (2014)**. This protocol is preferable to Cas9 genome editing for introduction of point mutations and small indels because it does not require introduction of synonymous PAM mutations that have the potential to affect RNA structure. The 58 bp deletion upstream of argB, 82 bp deletion in rph upstream of pyrE, 12 bp deletion in carB (at nt 2906), 132 bp deletion in carB (at nt 2986), and two stop codons in argC were introduced using Cas9-induced DNA cleavage and λ Red recombinase-mediated homology-directed repair with a linear DNA fragment. Sequences of the protospacers and mutation cassettes used for Cas9 genome editing procedures are listed in **Supplementary file 3** and **Supplementary file 4**.

The cells were first transformed with a helper plasmid (pAM053, **Supplementary file 1**) encoding cas9 under control of a weak constitutive promoter (pro1 from **Davis et al., 2011**), λ Red

recombinase genes (*exo*, *gam*, and *bet*) under control of a heat-inducible promoter, and a temperature-sensitive origin of replication (*Datta et al., 2006*). The cells were grown to an $OD_{600}$ of 0.2–0.4 at 30°C and then incubated at 42°C with shaking for 15 min to induce expression of the λ Red recombinase genes. The cells were immediately subjected to electroporation with 100 ng of a plasmid expressing a guide RNA targeting a 20-nucleotide sequence within the region targeted for deletion (*Supplementary file 1*, *Supplementary file 3*), and 450 ng of a linear homology repair template that encodes the new sequence with the desired deletion (*Supplementary file 4*). (Linear homology repair templates were amplified from genomic DNA of clones isolated during the evolution experiment or plasmids that contained the desired deletions and the PCR fragments were gel-purified. Primers used to generate the linear DNA mutation fragments are listed in *Supplementary file 2*.) The cells were allowed to recover at 30°C for 2–3 hr before being spread onto LB/chloramphenicol/ampicillin plates. Sanger sequencing confirmed that the surviving colonies contained the desired deletion. Individual colonies were cured of pAM053 and the guide RNA plasmids, both of which have temperature-sensitive origins of replication, by growth at 37°C.

## Laboratory evolution

Evolution of strain AM187 in Evolution Medium was carried out in eight replicate tubes in a custom turbidostat constructed as described by *Takahashi et al. (2015)*. To start the experiment, strain AM187 was grown to exponential phase ($OD_{600}$ = 0.7) in LB/kanamycin at 37°C. Cells were centrifuged at 4000 x *g* for 10 min at room temperature and resuspended in an equal volume of PBS. The suspended cells were washed twice more with PBS and resuspended in PBS. This suspension was used to inoculate all eight turbidostat chambers to give an initial $OD_{600}$ of 0.01 in 14 mL of Evolution Medium. The turbidostat was set to maintain an $OD_{650}$ of 0.4 by diluting individual cultures with an appropriate amount of fresh medium every 60 s.

A 3 mL portion of each population was collected every 2–3 days; 800 µl was used to make a 10% glycerol stock, which was then stored at −70°C. The remaining sample was pelleted for purification of genomic DNA using the Invitrogen PureLink Genomic DNA Mini Kit according to the manufacturer's protocol.

At several points during the evolution, the turbidostat was restarted due to a planned pause or an instrument malfunction. During a planned pause, the populations were subjected to centrifugation at 4000 x *g* for 10 min at room temperature and the pelleted cells were resuspended in 1.6 mL of Evolution Medium. Half of the resuspension was used to make a 10% glycerol stock for storage at −70°C, and the other half to purify genomic DNA. When the turbidostat was restarted, the frozen stock was thawed and the cells were collected by centrifugation at 16,000 x *g* for 1 min at room temperature. The pelleted cells were resuspended in 1 mL of PBS, washed, and resuspended in 500 µL of Evolution Medium. The entire resuspension was used to inoculate the appropriate chamber of the turbidostat. Sometimes the experiment had to be restarted from a frozen stock of a normal sample (as opposed to the entire population as just described), resulting in a more significant population bottleneck. In this case, the entire frozen stock was thawed and only 700 µL washed as described above to be used for the inoculation. The remaining 300 µL of the glycerol stock were re-stored at −70°C in case the frozen stock was needed for downstream analysis. The times at which the turbidostat failed and was restarted are indicated in *Figure 4—source data 1*. We always restarted the turbidostat with >$10^8$ cells (>5% of the culture) in order to preserve the diversity of the previous populations.

## Calculation of growth rate and generations during adaptation

The turbidostat takes an $OD_{650}$ reading every ~3 s and dilutes the cultures every 60 s. Thus, readings between dilutions can be used to calculate an average growth rate each day based on the following equation:

$$\bar{\mu} = \frac{\sum_i^n \frac{(N_{t1,i}/N_{t0,i})}{t_{1,i}-t_{0,i}}}{n} \tag{6}$$

where $\bar{\mu}$ is the average growth rate in $hr^{-1}$, $n$ is the number of independent growth rate calculations within a given 24 hr period, $N_{t0}$ is the $OD_{650}$ reading right after the dilution, $N_{t1}$ is the $OD_{650}$ reading

right before the next dilution, and $t_0$ and $t_1$ are the times at which the OD$_{650}$ was measured. The number of generations per day (g) was then calculated from using (**Equation 7**).

$$g = \frac{24\,h}{ln(2)/\bar{\mu}}$$ (7)

The R script used to calculate growth rate from turbidostat readings can be found in **Source code 1**.

## Measurement of *proA\** copy number

The copy number of *proA\** was determined by qPCR of purified population genomic DNA. *gyrB* and *icd*, which remained at a single copy in the genome throughout the adaptation experiment, were used as internal reference genes. The primer sets used for each gene are listed in **Supplementary file 2**. PowerSYBR Green PCR master mix (Thermo Scientific) was used according to the manufacturer's protocol. A standard curve using variable amounts of AM187 genomic DNA was run on every plate to calculate efficiencies for each primer set. Primer efficiencies were calculated with the following equation:

$$E_x = 10^{-(\frac{1}{m})}$$ (8)

where *E* is the efficiency of primer set *x*, and *m* is the slope of the plot of C$_t$ vs. starting quantity for the standard curve. *proA\** copy number was then calculated with the following equation (**Hellemans et al., 2007**):

$$n = \frac{E_{proA}^{\Delta C_{t,proA}}}{\sqrt{E_{gyrB}^{\Delta C_{t,gyrB}} \times E_{icd}^{\Delta C_{t,icd}}}}$$ (9)

where *n* is the *proA\** copy number, and $\Delta C_{t,x}$ is the difference in C$_t$'s measured during amplification of AM187 and sample genomic DNA with primer set *x*.

## Whole-genome sequencing

Libraries were prepared from purified population genomic DNA using a modified Illumina Nextera protocol and multiplexed onto a single run on an Illumina NextSeq500 to produce 151 bp paired-end reads (**Baym et al., 2015**), giving a 60–130-fold coverage of the AM187 genome. Reads were trimmed using BBtools v35.82 (DOE Joint Genome Institute) and mapped using *breseq* v0.32.1 using the polymorphism (mixed population) option (**Deatherage and Barrick, 2014**).

## Growth rate measurements

Growth rates for individual constructed strains were calculated from growth curves measured in quadruplicate. Overnight cultures were grown in LB at 37°C from glycerol stocks. Kanamycin (20 μg/mL) was added for strains in which *argC* had been replaced by *kan$^R$*. Ampicillin (100 μg/mL) was added for strains carrying the *argB* expression plasmid (pAM141, **Supplementary file 1**). Forty μL of each overnight culture was used to inoculate 4 mL of LB with appropriate antibiotics and the cultures were allowed to grow to mid-exponential phase (OD$_{600}$ 0.3–0.6) at 37°C with shaking. The cultures were subjected to centrifugation at 4000 x *g* for 10 min at room temperature and the pellets resuspended in an equal volume of PBS. The pellets were washed once more in PBS. The cells were diluted to an OD$_{600}$ of 0.001 in Evolution Medium and a 100 μL aliquot was loaded into each well of a 96-well plate. When *argB* was expressed from a low-copy plasmid carrying *amp$^r$* (**Figure 6B**), kanamycin was omitted and ampicillin was added to the medium. The plates were incubated in a Varioskan (Thermo Scientific) plate reader at 37°C with shaking every 5 min for 1 min. The absorbance at 600 nm was measured every 20 min for up to 200 hr. The baseline absorbance for each well (the average over several smoothed data points before growth) was subtracted from each point of the growth curve. Growth parameters (maximum specific growth, μ$_{max}$; lag time, λ; maximum growth, A$_{max}$) were estimated by non-linear regression using the modified Gompertz equation (**Zwietering et al., 1990**). Non-linear least-squares regression was performed in Excel using the Solver feature.

Growth rates were calculated for populations in the turbidostats during evolution and for individual strains in the plate reader. The growth rate of the parental strain AM187 is ~0.27 h$^{-1}$ in the plate reader (*Figure 5B*, *Figure 6B*, *Figure 7F*, *Figure 4—figure supplement 2*, *Figure 7—figure supplement 2*) and ~0.24 h$^{-1}$ in the turbidostat (*Figure 3*), so the growth rates of individual strains in the plate reader and turbidostat are similar.

## Fitness competition assay

Fitness competition assays were used in lieu of growth curves when growth rate differences between strains were expected to be small (*Figure 7G*). Overnight cultures of a reference strain containing a plasmid carrying *cfp* (pAM003, *Supplementary file 1*) and a test strain containing a plasmid carrying *yfp* (pAM142, *Supplementary file 1*) were grown in LB/ampicillin at 37°C from glycerol stocks. Forty μL of each overnight culture was inoculated into 4 mL of M9/0.2% glucose/ampicillin and the cultures were allowed to grow to mid-exponential phase (OD$_{600}$ 0.3–0.6) at 37°C with shaking. One mL of each culture was subjected to centrifugation at 10,000 x *g* for 1 min at room temperature and the pellets resuspended in an equal volume of PBS. The CFP- and YFP-labelled strains were mixed in equal parts to a final OD$_{600}$ of 0.01 in 25 mL of M9/0.2% glucose/ampicillin. Competition cultures were grown at 37°C with shaking and passaged into fresh M9/0.2% glucose/ampicillin at mid-log phase four times.

We used flow cytometry to count cells in initial and final cultures. The final cultures were diluted 100-fold in PBS prior to flow cytometry. Relative fitness was calculated by the following equation (*Dykhuizen, 1990*):

$$w = \frac{ln(R(t)/R(0))}{t} + 1 \tag{10}$$

Where *t* is the number of generations and *R* is the ratio of mutant to reference strain cell counts (YFP/CFP). All *w* values were normalized to the *w* value obtained for a competition between the CFP-containing reference strain and a YFP-containing reference strain to account for any fitness effects of expressing YFP versus CFP.

## Measurement of *argB* and *argH* gene expression by RT-qPCR

Overnight cultures were grown from glycerol stocks in LB/kanamycin at 37°C. Ten μL of each overnight culture was used to inoculate 4 mL of LB/kanamycin and the cultures were grown to mid-exponential phase (OD$_{600}$ 0.3–0.6) at 37°C with shaking. The cultures were centrifuged at 4000 x *g* for 10 min and pellets resuspended in equal volume PBS. Pellets were washed once more in PBS. The cells were diluted to an OD$_{600}$ of 0.001 in 4 mL of Evolution Medium and grown to an OD$_{600}$ of 0.2–0.3. Four 2 mL aliquots of culture were thoroughly mixed with 4 mL of RNAprotect Bacteria Reagent (Qiagen) and incubated at room temperature for 5 min before centrifugation at 4000 x *g* for 12 min at room temperature. Pellets were frozen in liquid N$_2$ and stored at −70°C.

RNA was purified using the Invitrogen PureLink RNA Mini Kit according to the manufacturer's protocol. The cell lysate produced during the PureLink protocol was homogenized using the QIAShredder column (Qiagen) prior to RNA purification. After RNA purification, each sample was treated with TURBO DNase (Invitrogen) according to the manufacturer's protocol. Reverse transcription (RT) was performed with 250–600 ng of RNA using SuperScript IV VILO (Invitrogen) master mix according to the manufacturer's protocol.

qPCR of cDNA was performed to measure the fold-change in expression of *argB* and *argH* in mutant strains compared to that in AM187. *hcaT* and *cysG* were used as reference genes (*Zhou et al., 2011*). The primer sets used for each gene are listed in *Supplementary file 2*. A standard curve using variable amounts of *E. coli* BW25113 genomic DNA was run to calculate the primer efficiencies for each primer set. Fold-changes in expression of *argB* and *argH* were calculated as described above for calculations of *proA** copy number.

## Measurement of ArgB and ArgH protein levels

Individual colonies were inoculated into four parallel 2 mL aliquots of LB. Kanamycin (20 μg/mL) was added for strains in which *argC* had been replaced by *kan*$^r$. Ampicillin (100 μg/mL) was added and kanamycin was omitted when *argB* was expressed from a low-copy plasmid (pAM141,

*Supplementary file 1*). The cultures were grown to mid-exponential phase at 37°C with shaking. One mL of each culture was subjected to centrifugation at 16,000 x *g* for 1 min at room temperature. The cell pellets were resuspended in 1 mL PBS and washed twice more in PBS before resuspension and dilution to an OD of 0.001 in 5 mL of Evolution Medium. Antibiotics were added as detailed above. Cultures were grown to an $OD_{600}$ of 0.1–0.3 at 37°C with shaking and then chilled on ice for 10 min before pelleting by centrifugation at 4000 x *g* at 4°C. Cell pellets were frozen in liquid $N_2$ and stored at −70°C.

Frozen cell pellets were thawed and lysed in 60 μL 50 mM Tris-HCl, pH 8.5, containing 4% (w/v) SDS, 10 mM tris(2-carboxyethylphosphine) (TCEP) and 40 mM chloroacetamide in a Bioruptor Pico sonication device (Diagenode) using 10 cycles of 30 s on, 30 s off, followed by boiling for 10 min, and then another 10 cycles in the Bioruptor. The lysates were subjected to centrifugation at 15,000 x *g* for 10 min at 20°C and protein concentrations in the supernatant were determined by tryptophan fluorescence (*Wiśniewski and Gaugaz, 2015*). Ten μL of each sample (3–6 μg protein/μL) was digested using the SP3 method (*Hughes et al., 2014*). Carboxylate-functionalized speedbeads (GE Life Sciences) were added to the lysates. Addition of acetonitrile to 80% (v/v) caused the proteins to bind to the beads. The beads were washed twice with 70% (v/v) ethanol and once with 100% acetonitrile. Protein was digested and eluted from the beads with 15 μL of 50 mM Tris buffer, pH 8.5, with 1 μg endoproteinase Lys-C (Wako) for 2 hr with shaking at 600 rpm at 37°C in a thermomixer (Eppendorf). One μg of trypsin (Pierce) was then added to the solution and incubated at 37°C overnight with shaking at 600 rpm. Beads were collected by centrifugation and then placed on a magnet to more reliably remove the elution buffer containing the digested peptides. The peptides were then desalted using an Oasis HLB cartridge (Waters) according to the manufacturer's instructions and dried in a speedvac.

Samples were suspended in 12 μL of 3% (v/v) acetonitrile/0.1% (v/v) trifluoroacetic acid and 0.5–1 μg of peptides were directly injected onto a C18 1.7 μm, 130 Å, 75 μm X 250 mm M-class column (Waters), using a Waters M-class UPLC. Peptides were eluted at 300 nL/minute using a gradient from 3% to 20% acetonitrile over 100 min into an Orbitrap Fusion mass spectrometer (Thermo Scientific). Precursor mass spectra (MS1) were acquired at a resolution of 120,000 from 380 to 1500 m/z with an AGC target of $2.0 \times 10^5$ and a maximum injection time of 50 ms. Dynamic exclusion was set for 20 s with a mass tolerance of + /− 10 ppm. Precursor peptide ion isolation width for MS2 fragment scans was 1.6 Da using the quadrupole, and the most intense ions were sequenced using Top Speed with a 3 s cycle time. All MS2 sequencing was performed using higher energy collision dissociation (HCD) at 35% collision energy and scanning in the linear ion trap. An AGC target of $1.0 \times 10^4$ and 35 s maximum injection time was used. Rawfiles were searched against the Uniprot *Escherichia coli* database using Maxquant version 1.6.1.0 with cysteine carbamidomethylation as a fixed modification. Methionine oxidation and protein N-terminal acetylation were searched as variable modifications. All peptides and proteins were thresholded at a 1% false discovery rate (FDR).

## Enzyme overexpression plasmids

*argB* and *proA* were amplified from the genome of *E. coli* BW25113 and *proA\** was amplified from the genome of AM187 using primers specified in *Supplementary file 2*. The amplified PCR fragments were ligated into a linearized pET-46 vector backbone by Gibson assembly (NEB) to make pAM028, pAM063, and pAM064, respectively (*Supplementary file 1*). A sequence encoding a 6xHis-tag followed by a 2xVal-linker was incorporated at the N-terminus of each protein. The *proA\*\** expression plasmid (pAM112) was constructed from pAM064 using the Q5 Site-Directed Mutagenesis Kit (NEB) and the primers listed in *Supplementary file 2*.

*argC* was cloned into a pTrcHisB vector backbone as described in *McLoughlin and Copley (2008)*. The final plasmid encodes ArgC with an N-terminal 6xHis-tag followed by a Gly-Met-Ala-Ser linker and with Met1 removed. *carAB* was amplified from the genome of AM187 and inserted into a PCR-amplified pCA24N vector backbone by Gibson assembly (NEB) to make pAM101. (PCR primers are listed in *Supplementary file 2*). The final construct included an N-terminal 6xHis-tag on CarA followed by a Thr-Asp-Pro-Ala-Leu-Arg-Ala linker. The Q5 Site-Directed Mutagenesis Kit (NEB) was used to generate mutant versions of *carB* in plasmids pAM102–109 using the primers listed in *Supplementary file 2*.

The *argD* and *argI* expression plasmids from the ASKA collection (*Kitagawa et al., 2005*) were used for expression of *N*-acetylornithine aminotransferase and ornithine transcarbamoylase,

respectively. These expression plasmids include a sequence encoding an N-terminal 6xHis-tag followed by a Thr-Asp-Pro-Ala-Leu-Arg-Ala linker upstream of each cloned gene.

The correct sequences for all constructs were confirmed by Sanger sequencing.

## Protein purification

Wild-type and variant ProAs were expressed in strain AM209 [BL21(DE3) *argC::kan^r proA::cat*] to avoid contamination with wild-type ProA and ArgC. Carbamoyl phosphate synthetase (CPS) consists of a stable complex between CarA and CarB. Thus, *carA* and wild-type or variant *carBs* were co-expressed on the same plasmid with a His-tag on CarA in strain AM267 (BL21 *carAB::kan^r*) to enable purification in the absence of wild-type CPS. Ornithine transcarbamoylase was also expressed in this strain. ArgB was expressed in BL21(DE3).

Enzymes were expressed and purified using the following protocol with minor variations. A small scraping from the glycerol stock of each expression strain was used to inoculate LB containing the antibiotics required for maintenance of each expression plasmid (*Supplementary file 1*). The cultures were grown overnight with shaking at 37°C. Overnight cultures were diluted 1:100 into 500 mL-2 L of LB containing the appropriate antibiotic and grown with shaking at 37°C. IPTG was added to a final concentration of 0.5 mM when the $OD_{600}$ reached 0.5–0.9. Growth was continued at 30°C for 5 hr with shaking. Cells were harvested by centrifugation at 5000 x *g* for 20 min at 4°C. Cell pellets were stored at −70°C until protein purification.

Frozen cell pellets were resuspended in 5x the cell pellet weight of ice-cold 20 mM sodium phosphate, pH 7.4, containing 300 mM NaCl and 10 mM imidazole. Fifty μL of protease inhibitor cocktail (Sigma-Aldrich, P8849) was added for each gram of cell pellet. Lysozyme was added to a final concentration of 0.2 mg/mL and the cells were lysed by probe sonication (20 s of sonication followed by 30 s on ice, repeated three times). Cell debris was removed by centrifugation at 18,000 x *g* for 20 min at 4°C. The soluble fraction was then loaded onto 1 mL or 3 mL HisPur Ni-NTA Spin Columns (Thermo Scientific) and His-tagged protein was purified according to the manufacturer's protocol. Bound protein was eluted with one column volume of 20 mM sodium phosphate, pH 7.4, containing 300 mM NaCl and increasing amounts of imidazole (100 mM, 250 mM, and finally 500 mM). Two separate elutions were performed with 500 mM imidazole. Fractions containing the protein of interest were pooled and dialyzed overnight against 6–12 L of exchange buffer at 4°C. (ProA and ArgC were dialyzed against 20 mM potassium phosphate, pH 7.5, containing 20 mM DTT. *N*-acetylornithine aminotransferase was dialyzed against 20 mM potassium phosphate, pH 7.5. CPS was dialyzed against 100 mM potassium phosphate, pH 7.6. Ornithine transcarbamoylase was dialyzed against 20 mM Tris-acetate, pH 7.5. ArgB was dialyzed against 10 mM Tris-HCl, pH 7.8.) Protein purity was assessed by SDS-PAGE and concentration measured using the Qubit protein assay kit with a Qubit 3.0 fluorometer (Invitrogen). Purified protein was stored at 4°C for short-term storage, and frozen in liquid nitrogen and stored at −70°C for long-term storage.

## GSA and NAGSA dehydrogenase assays

The native and neo-ArgC activities of ProA were assayed in the reverse direction (dehydrogenase reaction) because the lability of the forward substrates γ-glutamyl phosphate and *N*-acetylglutamyl phosphate makes them difficult to purify. The change in the dehydrogenase activity due to a mutation is proportional to the change in the reductase activity according to the Haldane relationship (*Haldane, 1930*; *McLoughlin and Copley, 2008*).

Assaying ProA's dehydrogenase activity using γ-glutamyl semialdehyde (GSA) and *N*-acetylglutamyl semialdehyde (NAGSA) as substrates is complicated by the equilibrium of GSA and NAGSA with their hydrated forms, as well as GSA's intramolecular cyclization to form pyrroline-5-carboxylate (P5C) (*Bearne and Wolfenden, 1995*; *Mezl and Knox, 1976*). In order to measure the concentration of the free aldehyde form of these substrates, we mixed 15 μM ProA or ArgC with 2 mM 'GSA' (including the hydrate and P5C) or 2 mM 'NAGSA' (including the hydrate), respectively, in a solution containing 100 mM potassium phosphate, pH 7.6, and 1 mM $NADP^+$ and measured the burst in NADPH production (*Khanal et al., 2015*). The concentrations of GSA+P5C+hydrate or NAGSA +hydrate were determined using the *o*-aminobenzaldehyde assay (*Albrecht et al., 1962*; *Mezl and Knox, 1976*). The absorbance at 340 nm due to formation of NADPH exhibited a burst followed by a linear phase that was followed for 60 s. We assume that the burst corresponds to reduction of the

free aldehyde form of GSA or NAGSA and the rate of the linear phase is determined by the conversion of the hydrate (and P5C in the case of GSA) to the free aldehyde. We calculated the magnitude of the burst by fitting either all of the data or the linear portion of the data to one of the following equations.

$$f(x) = mx + b \tag{11}$$

$$f(x) = mx + b(1 - e^{-x}) \tag{12}$$

where $x$ is time in seconds, $m$ is the slope of the linear phase, and $b$ is the magnitude of the burst and thus proportional to the starting concentration of the free aldehyde form of the substrate. In the case of the linear fit, only the linear portion of the $A_{340}$ data was used. *Equation 11* was used to calculated NAGSA free aldehyde concentration because the exponential equation did not fit the data well. *Equation 12* was used to calculated GSA free aldehyde concentration. We repeated the assay three times and averaged the magnitude of the burst to calculate free aldehyde concentrations for solutions of GSA and NAGSA (under these buffer and temperature conditions) of 4.5% and 4.2% of the total concentration of free aldehyde + hydrate (+ P5C for GSA), respectively.

GSA and NAGSA dehydrogenase activities were measured by monitoring the appearance of NADPH at 340 nm in reaction mixtures containing 100 mM potassium phosphate, pH 7.6, 1 mM NADP$^+$, varying concentrations of NAGSA or GSA, and catalytic amounts of ProA, ProA\*, and ProA\*\*. All kinetic measurements were done at 25°C. Values for $K_M$ refer to the concentration of the free aldehyde form of the substrate. An example R script used to calculate the Michaelis-Menten parameters can be found in *Source code 2* Table 2—source code 1.

## Assays for carbamoyl phosphate synthetase activity and allosteric regulation

Kinetic assays for carbamoyl phosphate synthetase (CPS) were carried out with minor modifications of the methods described in *Pierrat and Raushel (2002)*. The rate of ATP hydrolysis was measured at 37°C by coupling production of ADP to oxidation of NADH using pyruvate kinase, which converts ADP and PEP to ATP and pyruvate, and lactate dehydrogenase, which reduces pyruvate to lactate. Loss of NADH was monitored at 340 nm. Reaction mixtures consisted of 50 mM HEPES, pH 7.5, containing 10 mM MgCl$_2$, 100 mM KCl, 20 mM potassium bicarbonate, 10 mM L-glutamine, 1 mM PEP, 0.2 mM NADH, saturating amounts of pyruvate kinase and lactate dehydrogenase (Sigma-Aldrich, P0294), and varying amounts of ATP (0.01 to 8 mM). Reactions were initiated by adding CPS to a final concentration of 0.2 µM. The effects of UMP and ornithine were measured under the same reaction conditions but with a fixed ATP concentration of 0.2 mM and varying concentrations of either UMP or ornithine. Kinetic parameters were calculated from a nonlinear least squares regression of data for three technical replicates at each substrate concentration. Examples of R scripts used to calculate Michaelis-Menten parameters and parameters for allosteric regulation of CPS by UMP and ornithine can be found in *Source code 2* and *Source code 3*, respectively.

Carbamoyl phosphate production was measured with minor modifications of previously described procedures (*Snodgrass and Parry, 1969*; *Stapleton et al., 1996*). Formation of carbamoyl phosphate by CPS was coupled with formation of citrulline by ornithine transcarbamoylase; citrulline forms a yellow complex ($\varepsilon_{464}$ = 37800 M$^{-1}$ cm$^{-1}$; *Snodgrass and Parry, 1969*) when mixed with diacetyl monoxime and antipyrine. Reaction mixtures consisted of 50 mM HEPES, pH 7.5, 10 mM MgCl$_2$, 100 mM KCl, 20 mM potassium bicarbonate, 10 mM L-glutamine, 4 mM ATP, 10 mM L-ornithine, and 0.7 µM ornithine transcarbamoylase. Reactions (0.25 mL) were initiated by adding CPS at a final concentration of 0.2 µM. After incubation for 2.5 min at 37°C, reactions were quenched by addition of 1 mL of a solution consisting of 25% concentrated H$_2$SO$_4$, 25% H$_3$PO$_4$ (85%), 0.25% (w/v) ferric ammonium sulfate, and 0.37% (w/v) antipyrine, followed by addition of 0.5 mL of 0.4% (w/v) diacetyl monoxime/7.5% (w/v) NaCl. The quenched reaction mixtures were placed in a boiling water bath for 15 min before measurement of OD$_{464}$. Control reactions contained all components except CPS.

## RNA structure prediction

RNA secondary structures for *argB* mRNAs were predicted using CLC Main Workbench 8.1, which uses the *mfold* algorithm (*Mathews et al., 1999*). The entire intergenic region between *kan^r* and *argB* plus the first 33 nucleotides of *argB* were included in the structure prediction. The first 33 nucleotides were included because an mRNA-bound ribosome prevents another ribosome from binding to the mRNA until it has moved past the first 33 nucleotides (*Steitz, 1969*). Thus, at least the first 33 nucleotides are available for folding with the upstream region when a mRNA is being translated.

## Calculation of RNA folding times

Folding times for a 63-nucleotide region (30 nt downstream and 30 nt upstream of the *argB* start codon) surrounding the start codon of *argB* mRNAs were calculated using the Kinfold program (v1.3) from the ViennaRNA v2.4.11 package (*Wolfinger et al., 2004*). Kinfold utilizes a Monte Carlo algorithm to calculate the folding time of each RNA sequence to the lowest free energy structure. We simulated 500 folding trajectories for each structure.

## Calculation of sRNA-mRNA hybridization energy

Hybridization energies for sRNA-*argB* mRNA interactions were calculated using the IntaRNA algorithm, which predicts interacting regions between two RNA molecules by taking into account both the stability of sRNA-mRNA interactions and the accessibility of the interacting sequences (*Busch et al., 2008*; *Mann et al., 2017*; *Raden et al., 2018*; *Wright et al., 2014*). The 65 annotated non-coding RNAs in the *E. coli* BW25113 genome (GenBank accession number CP009273 *Grenier et al., 2014*) were used as query sRNAs. The RNA sequences encompassing the intergenic region between *kan^r* and *argB* through 33 bp downstream of the *argB* start codon were used as target mRNAs. Default parameters were applied with seven base pairs as the minimum size of the seed region.

## Data availability

The genome sequence of *E. coli* strain AM187 used in this study has been deposited to NCBI GenBank under accession number CP037857.1.

## Acknowledgements

We thank Craig Joy (University of Colorado Boulder, Physics Department) and Chris Takahashi (University of Washington) for help building the turbidostat. Publication of this article was funded by the University of Colorado Boulder Libraries Open Access Fund.

## Additional information

### Funding

| Funder | Grant reference number | Author |
| --- | --- | --- |
| National Aeronautics and Space Administration | NNA15BB04A | Vaughn S Cooper Shelley D Copley |
| Department of Defense | 13-34-RTA-FP-007 | William M Old |
| University of Colorado Boulder | Libraries Open Access Fund | Shelley D Copley |
| Defense Advanced Research Projects Agency | 13-34-RTA-FP-007 | William M Old |

The funders had no role in study design, data collection and interpretation, or the decision to submit the work for publication.

### Author contributions

Andrew B Morgenthaler, Conceptualization, Data curation, Software, Formal analysis, Validation, Investigation, Visualization, Methodology; Wallis R Kinney, Corinne M Walsh, Investigation, Acquired

and interpreted data; Christopher C Ebmeier, Data curation, Formal analysis, Investigation; Daniel J Snyder, Resources, Data curation, Funding acquisition, Investigation; Vaughn S Cooper, Resources, Data curation, Supervision, Funding acquisition, Investigation; William M Old, Shelley D Copley, Conceptualization, Resources, Formal analysis, Supervision, Funding acquisition, Investigation, Project administration

### Author ORCIDs
Andrew B Morgenthaler [iD] http://orcid.org/0000-0003-3822-0212
Christopher C Ebmeier [iD] http://orcid.org/0000-0001-7940-6190
Vaughn S Cooper [iD] http://orcid.org/0000-0001-7726-0765
Shelley D Copley [iD] https://orcid.org/0000-0001-9727-7919

### Decision letter and Author response
Decision letter https://doi.org/10.7554/eLife.53535.sa1
Author response https://doi.org/10.7554/eLife.53535.sa2

## Additional files
### Supplementary files
- Source code 1. R script used to calculate growth rate for each population from turbidostat $OD_{650}$ readings.
- Source code 2. Example R script used to calculate Michaelis-Menten parameters.
- Source code 3. Example R script used to calculate $K_d$ and $a$ for CPS.
- Supplementary file 1. Plasmids used in this study.
- Supplementary file 2. Primers used in this study.
- Supplementary file 3. Protospacers used for Cas9-mediated scarless genome editing.
- Supplementary file 4. Mutation cassettes used for Cas9-mediated genome editing.
- Transparent reporting form

### Data availability
The genome sequence of *E. coli* strain AM187 used in this study has been deposited to NCBI Gen-Bank under accession number CP037857.1. All other data generated or analyzed during this study are included in the manuscript and supporting files. Source code files have been provided for Figures 3 and 4 and Tables 2 and 3.

The following dataset was generated:

| Author(s) | Year | Dataset title | Dataset URL | Database and Identifier |
|---|---|---|---|---|
| Morgenthaler AB, Copley SD | 2019 | *Escherichia coli* BW25113 strain AM187, complete genome | https://www.ncbi.nlm.nih.gov/nuccore/CP037857.1/ | GenBank, CP037857.1 |

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
