## [Decision Letter]

**Acceptance summary:**

Pathways for the evolution of new genes are of broad interest. One route involves amplification (under selection) of a gene that has secondary promiscuous function followed by subsequent divergence. The idea has received experimental attention with focus on mutations leading to specialisation in each copy. Here, using a bacterial model system, Morgenthaler et al. show that mutations underpinning improvement in promiscuous function often occur in targets other than the duplicated genes. This surprising finding draws attention to a complex interplay of mutational events underpinning gene duplication and divergence.

**Decision letter after peer review:**

[Editors’ note: a previous version of this study was rejected after peer review, but the authors submitted for reconsideration. The first decision letter after peer review is shown below.]

Thank you for submitting your work entitled "Mutations that improve efficiency of a weak-link enzyme are rare compared to adaptive mutations elsewhere in the genome" for consideration by *eLife*. Your article has been reviewed by three peer reviewers, one of whom is a member of our Board of Reviewing Editors, and the evaluation has been overseen by a Senior Editor. The following individual involved in review of your submission has agreed to reveal their identity: Dan Andersson (Reviewer #2).

Our decision has been reached after consultation between the reviewers. Based on these discussions and the individual reviews below, we regret to inform you that your work will not be considered further for publication in *eLife*.

All three reviewers were enthusiastic about the study and agree that it adds usefully to an important topic. However all three felt that the work was not sufficiently complete to warrant publication at this time. Should you chose to follow suggestions of the reviewers and provide additional support for your hypotheses, then a fresh submission would be welcome.

*Reviewer #1:*

The paper is a nice addition to the work on the evolution of new gene function via mutations that take a promiscuous enzyme and make it better. The message is that while amplifications are a starting point, and may increase opportunity for additional mutations within the amplified region, mutations that improve fitness can also arise elsewhere in the genome. This comes as no real surprise and there are many studies showing the importance of compensatory mutation in many different contexts. Nonetheless, this is not to take from the importance of tracking down the mutations in this particular context and understanding their contributions to the refinement of promiscuous function.

The bulk of the paper attempts to shed light on a handful of mutations (out of a great many) in candidate genes. The approach is largely biochemical. The findings are used to support a particular explanation, but where I feel the paper falls short is its failure to consider alternate possibilities (and to test them). Or put another way, the model presented by the authors needs experimental validation. This is particularly relevant to the *carB* side of the story, where at the very least the work needs to be backed by reconstruction of mutants in the ancestral background. Without this it is impossible to know whether the mutations are responsible for fitness increase and whether these depend on other mutations. And thus whether the model provided by the authors is correct. And indeed, whether explanation focussing on the arginine pathways is correct.

*Reviewer #2:*

Overall, I think the manuscript is interesting and the work well executed. The authors utilize a number of different experiments to support their conclusions. They also put in a considerable amount of effort to address/explain any potential discrepancies or inconsistencies in their data. However, even though the study presents some interesting conclusions, in the current manner in which the manuscript is written I think the authors undersell their work (or don't articulate its significance well) and the focus of the manuscript gets lost in the Results section.

Another aspect is whether or not, given the experimental design, these were the expected results. The initial mutations (promoter mutations and amplifications) that increase the expression of *proA** and were observed in the previous study would result in reduced selection for improving the enzyme's catalytic activity. Hence selection is more likely to proceed through other targets simply because of this reduced selective pressure, and a larger target size for alternate mutations. In other words, is the observation in this study an outcome of limited adaptive routes for improving catalytic activity, or an outcome of reduced selective pressure because of how the experiment was set-up. It would be nice if the authors could bring this up either in the Introduction or in the Discussion.

1) The Abstract could benefit from more elaboration on the key findings – the authors should briefly mention the three mechanisms of adaptive mutations they find and how everything relates back to proA and ultimately ties into arginine synthesis. They should better emphasize the significance and impact of their work. The same holds true for the second-to-last paragraph in the Introduction.

2) The main focus/purpose of the paper is lost in the Results section. It would help if the authors relate their findings back to proA/arginine pathway in each subsection, instead of doing so only in the Discussion.

3) Introduction, last paragraph: Should be rephrased. The mutations are still presumably affecting the same phenotype of arginine synthesis. Hence fitness is still increasing by the same mechanism.

4) It might be useful to include a schematic for the construction of the DELargC *proA*-yfp* strain in a main text or supplementary figure for added clarity in the Results section where strain construction is described (subsection “Growth rate of ∆*argC proA* E. coli* increased 3-fold within a few hundred generations of adaptation in M9/glucose/proline”) and/or in the corresponding Materials and methods section.

5) The "Mutations outside of *proA** improved fitness" subsection in the Results should be renamed. The current title of this subsection is somewhat misleading, as this subsection only focuses on the media-adapted mutations that do not directly pertain to *proA** and arginine synthesis. The mutations "outside of *proA**" that are truly important for improved fitness already have their own dedicated subsections.

6) "Mutations upstream of *argB* increase ArgB abundance": "The thermodynamic stability of this region is clearly not the only factor responsible for the effects of the mutations upstream of *argB*." and

"The effects of the -94 A➝G and -22 C➝A mutations, however, cannot be explained by either of these mechanisms." Can the authors suggest any other possible mechanisms to explain these observations?

7) "Mutations in *carB* either increase activity or impact allosteric regulation":

As mentioned above, relating these enzyme kinetics and regulation data back to the proA/arginine synthesis pathway would improve the focus and flow of the paper.

8) "Discussion": The lengthy description of the previously found *proBA* operon promoter mutations seems unnecessary. Maybe briefly mention the rationale behind using the M2 promoter mutation in the Results section where strain construction is described. Additionally, any explanation for why the M2 mutation was chosen over the M1 or M3 promoter mutations is currently missing in the manuscript.

9) There is a discord between the increase in fitness of the evolving populations (Figure 3) and the change in copy number of *proA** (Figure 4). Essentially the authors observe an increase in fitness (2-3 fold) in the first 100 generations in all the populations, but there is no change in *proA** copy number during this period. The authors should address this in the Discussion.

10) Subsection “Laboratory adaptation”, last paragraph: It would be good to report which cultures had to be restarted with this kind of a bottleneck, and whether that affected the adaptive mutations observed.

11) Subsection “Calculation of growth rate and generations during adaptation” and Equation 6: The terminologies used in the description and in the equation do not match.

12) For some of the experiments, growth rates are measured from growth curves using a plate reader, and different parameters are then calculated (subsection “Growth rate measurements”). However, it is not clear how these correspond to parameters under selection in a turbidostat, which was used for the evolution experiments. This comparison is important and should be made clear in the text and figures.

*Reviewer #3:*

The authors have developed a potentially very useful approach for testing the IAD model, where a promiscuous activity in a ProA mutant can complement an ArgC deletion. As expected based on previous experiments (Kershner et al., 2016) all eight populations increase the copy number of the weak-link enzyme, but in only one population a mutation in ProA is found. Instead mutations in a range of other genes are found, often repeatedly in the same genes.

The article is clearly written and methodologically sound and tests of the IAD model could be of interest for a wide audience. However I do not fully agree with the interpretation of some experiments and the not all of the authors' conclusions are supported by data:

The authors introduce a weak-link enzyme, but there are also other artificially introduced potential weak-links and these are what is targeted by mutation after the main fitness increase by the amplifications. For example the *rph-pyrE* mutations correct a known defect in pyrimidine biosynthesis in this particular strain.

The insertion of the kanamycin resistance cassette upstream in ArgC is likely to disrupt expression of ArgB and the mutations seen in the intergenic region might simply compensate for this. That they are relevant outside this artificial genetic context would have to be shown in for example an ArgC mutant with a smaller inactivating in-frame deletion of ArgC. Based on the genetic context in the wild type it is possible that there is some kind of translational link between ArgC and ArgB (just 7 bp intergenic in the wild type). This would be supported by the big deletion bringing the *kan^r^* ORF and ArgB closer that results in the largest increase in growth rate.

The computational analysis of effects of the mutations on mRNA structure is not relevant for a native context but mainly looks at effects caused by the strong secondary structure introduced by the FRT site in the *kan^r^* cassette inserted upstream of ArgB. By including selection to keep kanamycin resistance this will also constrain the types of possible mutations (for example deletion to regain its native promoter upstream of ArgC) thereby biasing mutations to the small artificial intergenic region.

The effect of the *carB* mutations are well characterized at the biochemical level, but the authors do not show how this increase fitness or even that the mutations increase growth rate when reconstructed in a wild type background. The hypothesis that this is due to an increase in carbamoyl phosphate synthase activity that potentiates flux through the arginine pathway is not supported by data. The authors might want to discuss alternative mechanisms such as that a decrease in ornithine concentration reduce the activation of CarB to such level that not enough carbamoyl phosphate is produced for pyrimidine synthesis (which is also crippled by the *rph-pyrE* mutation).

Abstract: "We have identified the mechanisms by which three classes of adaptive mutations increase fitness" No mechanisms are actually shown even though some reasonable interpretations are given.

There is no data supporting that the mutations would produce a cost in another context and therefore statements like "These changes could be detrimental requiring reversion or compensation" "often at a cost to a previously well-evolved function" should be avoided. The well-evolved functions seem to already be disrupted by the *rph-pyrE* mutation and the insertion of the kanamycin cassette.

Similarly: "While a constitutively active CPS is beneficial in the short term to improve arginine synthesis, it will likely be detrimental once arginine production no longer limits growth." This could be tested relatively easy experimentally, without such data it is merely a (very reasonable) hypothesis.

---

## [Author Response]

[Editors’ note: the author responses to the first round of peer review follow.]

Reviewer #1:The paper is a nice addition to the work on the evolution of new gene function via mutations that take a promiscuous enzyme and make it better. The message is that while amplifications are a starting point, and may increase opportunity for additional mutations within the amplified region, mutations that improve fitness can also arise elsewhere in the genome. This comes as no real surprise and there are many studies showing the importance of compensatory mutation in many different contexts. Nonetheless, this is not to take from the importance of tracking down the mutations in this particular context and understanding their contributions to the refinement of promiscuous function.The bulk of the paper attempts to shed light on a handful of mutations (out of a great many) in candidate genes. The approach is largely biochemical. The findings are used to support a particular explanation, but where I feel the paper falls short is its failure to consider alternate possibilities (and to test them). Or put another way, the model presented by the authors needs experimental validation. This is particularly relevant to the carB side of the story, where at the very least the work needs to be backed by reconstruction of mutants in the ancestral background. Without this it is impossible to know whether the mutations are responsible for fitness increase and whether these depend on other mutations. And thus whether the model provided by the authors is correct. And indeed, whether explanation focussing on the arginine pathways is correct.

We have added new experiments to address these concerns. 1) We have shown that the *carB* mutants do in fact increase fitness when introduced into the parental strain (Figure 7F). 2) The first mutations acquired during the evolution experiment were between *rph* and *pyrE*. These mutations correct a known defect in pyrimidine synthesis in *E. coli* K12 strains. We have shown that pyrimidine synthesis is no longer limiting after acquisition of the *rph-pyrE* mutation. However, poor arginine synthesis still limits growth (Figure 7—figure supplement 2). Thus, the *carB* mutations occur in a background in which there is selective pressure for improvement of arginine synthesis.

Reviewer #2:Overall, I think the manuscript is interesting and the work well executed. The authors utilize a number of different experiments to support their conclusions. They also put in a considerable amount of effort to address/explain any potential discrepancies or inconsistencies in their data. However, even though the study presents some interesting conclusions, in the current manner in which the manuscript is written I think the authors undersell their work (or don't articulate its significance well) and the focus of the manuscript gets lost in the Results section.

We have modified the text and figures to better focus the Results section on the impact of mutations on the arginine synthesis pathway. We have modified Figure 2 to show the connections between the arginine, proline, and pyrimidine synthesis pathways for reference throughout the Results and Discussion sections.

We have provided more context for why the *argB* mutations may be beneficial in the *argB* Results subsection, " Mutations upstream of *argB* increase ArgB abundance”. We have provided more context for why the *carB* mutations may be beneficial in the *carB* Results subsection, “Mutations in *carB* either increase activity or impact allosteric regulation”, and in a modified Figure 7B. We have modified Figure 8A to indicate the effects of each of the investigated mutations and added Figure 8B (the evolutionary trajectories of the evolved strains) to better summarize how the mutations impact fitness.

Another aspect is whether or not, given the experimental design, these were the expected results. The initial mutations (promoter mutations and amplifications) that increase the expression of proA* and were observed in the previous study would result in reduced selection for improving the enzyme's catalytic activity. Hence selection is more likely to proceed through other targets simply because of this reduced selective pressure, and a larger target size for alternate mutations. In other words, is the observation in this study an outcome of limited adaptive routes for improving catalytic activity, or an outcome of reduced selective pressure because of how the experiment was set-up. It would be nice if the authors could bring this up either in the Introduction or in the Discussion.

Although the selective pressure is reduced by the promoter mutation and gene amplification, the observation that each strain acquired 6-20 copies of a segment of the genome ranging between 5 and 164 kb demonstrates that there is still strong selective

pressure for improvements in arginine synthesis – by any mechanism. The observation of the preponderance of alternate mutations is very likely due to the larger target size. We have addressed this point in the tenth paragraph of the Discussion.

1) The Abstract could benefit from more elaboration on the key findings – the authors should briefly mention the three mechanisms of adaptive mutations they find and how everything relates back to proA and ultimately ties into arginine synthesis. They should better emphasize the significance and impact of their work. The same holds true for the second-to-last paragraph in the Introduction.

We have reworded the Abstract and the second-to-last paragraph in the Introduction as requested.

2) The main focus/purpose of the paper is lost in the Results section. It would help if the authors relate their findings back to proA/arginine pathway in each subsection, instead of doing so only in the Discussion.

See response to reviewer 2’s first comment above.

3) Introduction, last paragraph: Should be rephrased. The mutations are still presumably affecting the same phenotype of arginine synthesis. Hence fitness is still increasing by the same mechanism.

In this concluding paragraph of the Introduction, we are making a general point about evolution when a weak-link enzyme limits fitness, rather than a specific point about this system. We prefer to leave the statement as is.

4) It might be useful to include a schematic for the construction of the DELargC proA*-yfp strain in a main text or supplementary figure for added clarity in the Results section where strain construction is described (subsection “Growth rate of ∆argC proA* *E. coli* increased 3-fold within a few hundred generations of adaptation in M9/glucose/proline”) and/or in the corresponding Materials and methods section.

Done (Figure 2—figure supplement 1).

5) The "Mutations outside of proA* improved fitness" subsection in the Results should be renamed. The current title of this subsection is somewhat misleading, as this subsection only focuses on the media-adapted mutations that do not directly pertain to proA* and arginine synthesis. The mutations "outside of proA*" that are truly important for improved fitness already have their own dedicated subsections.

Done (subsection “Some prevalent mutations in the evolved clones are not related to improved arginine synthesis”).

6) "Mutations upstream of argB increase ArgB abundance": "The thermodynamic stability of this region is clearly not the only factor responsible for the effects of the mutations upstream of argB." and"The effects of the -94 A➝G and -22 C➝A mutations, however, cannot be explained by either of these mechanisms." Can the authors suggest any other possible mechanisms to explain these observations?

One other possibility is that the mutations upstream of *argB* might affect binding of an sRNA. We have added a discussion of this possibility to the text (see subsection “Mutations upstream of *argB* increase ArgB abundance” and Figure 6—figure supplement 4).

7) "Mutations in carB either increase activity or impact allosteric regulation":As mentioned above, relating these enzyme kinetics and regulation data back to the proA/arginine synthesis pathway would improve the focus and flow of the paper.

Done (subsection “Mutations in *carB* either increase activity or impact allosteric regulation”, Discussion).

8) "Discussion": The lengthy description of the previously found proBA operon promoter mutations seems unnecessary. Maybe briefly mention the rationale behind using the M2 promoter mutation in the Results section where strain construction is described. Additionally, any explanation for why the M2 mutation was chosen over the M1 or M3 promoter mutations is currently missing in the manuscript.

We have removed this description of the previously found *proBA* promoter mutations from the Discussion. The M3 mutation was not in the promoter of the *proBA* operon; we have clarified this in the text. There was no particular reason for choosing the M2 promoter mutation. We could have initiated the evolution experiment with a strain carrying the M1 mutation.

9) There is a discord between the increase in fitness of the evolving populations (Figure 3) and the change in copy number of proA* (Figure 4). Essentially the authors observe an increase in fitness (2-3 fold) in the first 100 generations in all the populations, but there is no change in proA* copy number during this period. The authors should address this in the Discussion.

The initial increase in fitness is caused by the mutations between *rph* and *pyrE*, which occur within the first 100 generations prior to *proA** amplification. We have modified the text and figures to emphasize this point (see subsection “Some prevalent mutations in the evolved clones are not related to improved arginine synthesis”, Figure 4—figure supplement 2, Figure 8B).

10) Subsection “Laboratory adaptation”, last paragraph: It would be good to report which cultures had to be restarted with this kind of a bottleneck, and whether that affected the adaptive mutations observed.

We have created a table with all the times the cultures had to be restarted in Figure 4—source data 1, sheet 4. We have also addressed this potential bottleneck in the subsection “Laboratory evolution”.

11) Subsection “Calculation of growth rate and generations during adaptation” and Equation 6: The terminologies used in the description and in the equation do not match.

Fixed (subsection “Calculation of growth rate and generations during adaptation”).

12) For some of the experiments, growth rates are measured from growth curves using a plate reader, and different parameters are then calculated (subsection “Growth rate measurements”). However, it is not clear how these correspond to parameters under selection in a turbidostat, which was used for the evolution experiments. This comparison is important and should be made clear in the text and figures.

This comparison is not particularly relevant because the growth rates in the turbidostat are for populations, while the growth rates measured in the plate reader are for individual clones constructed by introducing particular mutations into the parental or wild-type background. However, the growth rate of the parental strain AM187 is 0.27 hr-1 in the plate reader (Figure 5B, Figure 6B, Figure 7F, Figure 4—figure supplement 2) and about 0.24 h-1 in the turbidostat (Figure 3), so the calculated growth rates of individual clones in the plate reader and turbidostat are similar. This information has been added to the subsection “Growth rate measurements”.

Reviewer #3:The authors have developed a potentially very useful approach for testing the IAD model, where a promiscuous activity in a ProA mutant can complement an ArgC deletion. As expected based on previous experiments (Kershner et al., 2016) all eight populations increase the copy number of the weak-link enzyme, but in only one population a mutation in ProA is found. Instead mutations in a range of other genes are found, often repeatedly in the same genes.The article is clearly written and methodologically sound and tests of the IAD model could be of interest for a wide audience. However I do not fully agree with the interpretation of some experiments and the not all of the authors' conclusions are supported by data:The authors introduce a weak-link enzyme, but there are also other artificially introduced potential weak-links and these are what is targeted by mutation after the main fitness increase by the amplifications. For example the rph-pyrE mutations correct a known defect in pyrimidine biosynthesis in this particular strain.The insertion of the kanamycin resistance cassette upstream in ArgC is likely to disrupt expression of ArgB and the mutations seen in the intergenic region might simply compensate for this. That they are relevant outside this artificial genetic context would have to be shown in for example an ArgC mutant with a smaller inactivating in-frame deletion of ArgC. Based on the genetic context in the wild type it is possible that there is some kind of translational link between ArgC and ArgB (just 7 bp intergenic in the wild type). This would be supported by the big deletion bringing the kan^r^ ORF and ArgB closer that results in the largest increase in growth rate.

We have addressed this valid concern by measuring protein levels in AM187 and a comparable strain that lacks ArgC due to introduction of two stop codons by label-free proteomics. ArgB levels are indeed diminished by 2.3-fold in AM187 (Figure 6—figure supplement 1). However, growth rate of AM187 is increased by mutations that increase ArgB levels by up to 8-fold and by overexpression of ArgB up to 25-fold, demonstrating that the beneficial effect of the mutations we observed is not simply due to

compensation for the 2.3-fold decrease in ArgB caused by replacement of *argC* with *kan^r^*. This new experiment is described in the fourth paragraph of the subsection “Mutations upstream of *argB* increase ArgB abundance” and the data are shown in Figure 6B.

The computational analysis of effects of the mutations on mRNA structure is not relevant for a native context but mainly looks at effects caused by the strong secondary structure introduced by the FRT site in the kan^r^ cassette inserted upstream of ArgB. By including selection to keep kanamycin resistance this will also constrain the types of possible mutations (for example deletion to regain its native promoter upstream of ArgC) thereby biasing mutations to the small artificial intergenic region.

We agree that the effects of the mutations on mRNA structure are not relevant for a native context. However, they are relevant for the context in which the experimental evolution was carried out. A deletion to regain the native promoter upstream of *argC* would not be needed because insertion of *kan^r^* does not prevent utilization of the native promoter. The native promoter is regulated by the repressor ArgR, and would be expected to be active unless arginine levels are adequate. Thus, insertion of *kan^r^* does not interfere with transcription of the operon. Further, the secondary structure introduced by the FRT site (the large stem loop from nucleotide 25-61 in Figure 6—figure supplement 2) does not seem to be driving the mutations, as this region is affected in only two of the mutants (the Δ58 bp and Δ51 bp mutants). Most of the mutations do not disrupt the secondary structure caused by the FRT site.

The effect of the carB mutations are well characterized at the biochemical level, but the authors do not show how this increase fitness or even that the mutations increase growth rate when reconstructed in a wild type background. The hypothesis that this is due to an increase in carbamoyl phosphate synthase activity that potentiates flux through the arginine pathway is not supported by data. The authors might want to discuss alternative mechanisms such as that a decrease in ornithine concentration reduce the activation of CarB to such level that not enough carbamoyl phosphate is produced for pyrimidine synthesis (which is also crippled by the rph-pyrE mutation).

We are not entirely sure what the reviewer means by “even that the mutations increase growth rate when reconstructed in a wild-type background”. We suspect that the reviewer meant that we had not shown that the mutations increase growth rate in the parental background. We have added data showing that three of four of the *carB* mutations do indeed increase growth rate of the parental strain. A reasonable explanation for the lack of growth increase with the fourth is also included in the paper. (See subsection “Mutations in *carB* either increase activity or impact allosteric regulation”, seventh paragraph and Figure 7F.)

At the reviewer’s suggestion, we considered the possibility that the *carB* mutations might be beneficial because they improve pyrimidine synthesis rather than arginine synthesis. After acquisition of the *rph-pyrE* mutation, addition of uracil to the medium does not improve growth rate, although it does improve growth rate of the parental strain, which has a known defect in pyrimidine synthesis (see Figure 7—figure supplement 2). Thus, there is no longer selective pressure to improve pyrimidine synthesis after the acquisition of the *rph-pyrE* mutation. However, poor arginine synthesis still limits growth; addition of arginine restored growth to wild-type levels. Since the *carB* mutations occurred after the *rph-pyrE* mutations, we can conclude that these mutations are beneficial because they improve arginine synthesis. We have added this new experiment to the text in Figure 7—figure supplement 2 and the tenth paragraph of the subsection “Mutations in *carB* either increase activity or impact allosteric regulation”.

Abstract: "We have identified the mechanisms by which three classes of adaptive mutations increase fitness" No mechanisms are actually shown even though some reasonable interpretations are given.

By mechanisms we meant the increase in ProA* concentration produced by gene amplification, the increased activity of CarAB, and the increased expression of ArgB.

There is no data supporting that the mutations would produce a cost in another context and therefore statements like "These changes could be detrimental requiring reversion or compensation" "often at a cost to a previously well-evolved function" should be avoided. The well-evolved functions seem to already be disrupted by the rph-pyrE mutation and the insertion of the kanamycin cassette.Similarly: "While a constitutively active CPS is beneficial in the short term to improve arginine synthesis, it will likely be detrimental once arginine production no longer limits growth." This could be tested relatively easy experimentally, without such data it is merely a (very reasonable) hypothesis.

We have addressed this concern by introducing four of the *carB* mutations into the genome of wild-type *E. coli* containing the *rph-pyrE* mutation to simulate the situation where arginine production no longer limits growth. Competitive fitness assays showed that three mutations that alter allosteric regulation of CarB did in fact decrease fitness. Interestingly, one mutation increased growth rate in this background. These results are shown in Figure 7G and discussed in the Discussion.